# When does knowledge distillation improve robustness?

## Abstract

Knowledge distillation, typically employed to condense a large 'teacher' network into a smaller 'student' network, has been found to also effectively transfer adversarial robustness into mobile-friendly students. In this study, however, we show that knowledge distillation between large models can also be used to purely enhance adversarial robustness. Specifically, we present a thorough analysis of different robust knowledge distillation (RKD) techniques with the aim to provide general guidelines to improve the adversarial performance of a student model. Our ablations demonstrate the significance of early stopping, model ensembling, label mixing, and the use of weakly adversarially trained teachers as keys to maximize a student's performance; but we also find that matching the student and teacher in adversarial regions is beneficial in some settings. We thus introduce a new adversarial knowledge distillation loss (AKD) which matches the student's and teacher's output on adversarial examples, to study when it can be beneficial in the context of RKD. Finally, we use our insights to enhance the state-of-the-art robust models and find that while our proposed guidelines can complement and improve them, the main achievable performance benefits still depend on the quantity and quality of the training data used.

## 1 Introduction

Knowledge distillation (KD) (Hinton et al., 2015) is a common technique used to compress neural networks to be deployed on computationally constrained devices (Vanhoucke et al., 2011; Jaderberg et al., 2014; Sandler et al., 2018). By matching the output of a small lightweight model, or *student*, with that of a large neural network, or *teacher*, KD manages to transfer the performance of state-of-the-art networks to small architectures (Beyer et al., 2022). Surprisingly, it has recently been discovered that, even when both models have the same capacity, sometimes the student can perform better than its teacher (Furlanello et al., 2018; Bagherinezhad et al., 2018; Dong et al., 2019). This effect has been attributed to the existence of additional hidden information about the learned representations of the teacher in its outputs (e.g., similarity between classes) that the student uses to learn more effectively the task and avoid memorization pitfalls. This information is usually referred to as *dark knowledge* (Hinton et al., 2015; Furlanello et al., 2018; Yang et al., 2018).

However, accuracy may not be the only objective, and robustness has also become an important property for neural networks (Ortiz-Jiménez et al., 2021). While KD is effective at transferring performance, its classical form does not effectively transfer the adversarial robustness of the teacher model (Carlini & Wagner, 2017; Goldblum et al., 2020). Robust Knowledge Distillation (RKD) methods have thus been proposed to create lightweight robust student models that learn from larger robust teacher models (Goldblum et al., 2020; Zi et al., 2021; Shao et al., 2021). Nevertheless, to the best of our knowledge, using RKD to obtain a student model that outperforms its teacher in terms of robustness has not been studied thoroughly before, so it still remains an open question to know if RKD can be used between teacher and student models of the same capacity.

In this work, we present a thorough analysis to gauge the ability of RKD in improving model's robustness. To do so, we explore different design factors of RKD to find the most effective way to boost robustness. First, we study the use of early stopping when training the teacher model, which does not only increase the student model robustness but also reduces the computational cost of training the teacher model. Second, we analyze the effect of label mixing, in which the outputs of the student are matched with a linear combination of the teacher outputs and the true labels, and show that it increases the performance of the student model

Table 1: Knowledge distillation using teacher early stopping and our proposed label mixing can boost the clean and the robust performance of the student model. We observe that depending on how the teacher is trained, either the clean or the robust performance of the student model are maximized. We study the effect with no additional synthetic data, and 1 million examples generated from either Denoising Diffusion Probabilistic Models (DDPM) or Elucidating Diffusion Models (EDM). Results for our proposed $AKD_+$ loss function (Equation 7), tested with AutoAttack (Croce & Hein, 2020) $l_\infty$ adversarial perturbations of size $\varepsilon = 8/255$. ST: standard training. AT: adversarial training (Madry et al., 2018). LAT: low-epsilon adversarial training ($\varepsilon = 2/255$). AT+: adversarial training using improvements from (Rebuffi et al., 2021)

| Dataset | Student | Teacher | Student accuracy | Student robust acc. |
|---|---|---|---|---|
| CIFAR10 | ResNet-18 (AT) | No teacher | $85.54_{\pm.18}$ | $46.33_{\pm.05}$ |
| | ResNet-18 | ResNet-18 (LAT) | $\mathbf{87.37}_{\pm.56}$ | $46.71_{\pm.16}$ |
| | ResNet-18 | ResNet-18 (AT) | $84.11_{\pm.10}$ | $\mathbf{49.15}_{\pm.22}$ |
| +DDPM | WideResNet-28 (AT+) | No teacher | $87.95_{\pm.18}$ | $58.76_{\pm.72}$ |
| | WideResNet-28 | WideResNet-28 (AT+) | $\mathbf{88.35}_{\pm.14}$ | $\mathbf{60.78}_{\pm.43}$ |
| +EDM | WideResNet-28 (AT+) | No teacher | $\mathbf{92.84}_{\pm.04}$ | $61.69_{\pm.15}$ |
| | WideResNet-28 | WideResNet-28 (AT+) | $92.67_{\pm.07}$ | $\mathbf{62.08}_{\pm.01}$ |
| CIFAR100 | ResNet-18 (AT) | No teacher | $60.29_{\pm.08}$ | $22.38_{\pm.32}$ |
| | ResNet-18 | ResNet-18 (LAT) | $\mathbf{62.90}_{\pm.23}$ | $21.40_{\pm.10}$ |
| | ResNet-18 | ResNet-18 (AT) | $58.41_{\pm.09}$ | $\mathbf{24.53}_{\pm.11}$ |
| +DDPM | WideResNet-28 (AT+) | No teacher | $\mathbf{61.96}_{\pm.23}$ | $29.37_{\pm.35}$ |
| | WideResNet-28 | WideResNet-28 (AT+) | $\mathbf{62.13}_{\pm.09}$ | $\mathbf{30.48}_{\pm.50}$ |
| +EDM | WideResNet-28 (AT+) | No teacher | $\mathbf{71.53}_{\pm.24}$ | $32.60_{\pm.30}$ |
| | WideResNet-28 | WideResNet-28 (AT+) | $70.84_{\pm.07}$ | $\mathbf{34.35}_{\pm.18}$ |

at no additional cost. We further provide a rigorous ablation study outlining the influence that differently trained teachers have in the student model accuracy and robustness. Finally, we provide a new loss function, called adversarial knowledge distillation (AKD), which generalizes adversarial training for distillation and incorporates our label mixing proposal in order to train the student model. Compared to other RKD methods, we find that AKD can be especially effective in increasing the student model robustness when there is a large quantity of data, as it ensures the student model outputs exactly match the teacher outputs in the adversarial regions.

Overall, the main discoveries of our work are:

- RKD can be used to boost the performance of adversarially trained robust models as a simple plugin method to improve robustness and/or clean accuracy of state-of-the-art robust models (Table 1).

- Early-stopped teachers increase robustness on the student model. The stopping point for pretraining that maximizes post-distillation performance, is a network- and dataset-specific hyperparameter.

- Linearly mixing clean labels with the adversarial outputs of a pretrained teacher improves performance by leveraging our trust in the teacher predictions and the quality of the training data.

- Distilling from an adversarially trained teacher can increase the student robustness, while distilling from a standardly train model increases clean accuracy. Surprisingly, models trained adversarially using small $\varepsilon$ attacks can also improve clean accuracy while transferring some robustness.

- Distilling knowledge from an ensemble of models can further improve performance, even when the individual models are only weakly robust. Combined ensembles of standard and robust teacher models can be used to achieve a wider range of clean and robust performance trade-offs.

We provide deep analysis and solid empirical evidence for all of these discoveries, and show their wide applicability. This further motivates the use of RKD, not only to compress big networks, but to also increase their robustness performance. We eventually propose to follow simple guidelines to maximize the benefits of RKD, while reducing the added computational cost of training the teacher network.

## 2 Robust Knowledge Distillation

Knowledge Distillation (KD) consists on training a student model $f_S : \mathbb{R}^d \to \mathbb{R}^c$ to mimic the outputs of a teacher model $f_T : \mathbb{R}^d \to \mathbb{R}^c$. To that end, most KD methods (Hinton et al., 2015; Romero et al., 2014; Zagoruyko & Komodakis, 2016a; Chebotar & Waters, 2016) append a matching function to the cross-entropy loss which encourages the final representations of $f_S$ and $f_T$ to be close at every sample $\boldsymbol{x} \in \mathbb{R}^d$ with associated label $y \in \{0, 1\}^c$. In other words, Clean Knowledge Distillation (CKD) methods optimize the loss

$$\text{CKD}(\boldsymbol{x}, y) = (1 - \lambda)\text{CE}(f_S(\boldsymbol{x}), y) + \lambda\text{KL}(f_S(\boldsymbol{x}), f_T(\boldsymbol{x})) \tag{1}$$

where $f_S(\boldsymbol{x})$ and $f_T(\boldsymbol{x})$ are the probability outputs of the student and the teacher models, CE denotes the cross-entropy loss, KL the Kullback-Leibler divergence, and $\lambda \in [0, 1]$ is the strength of the regularization. While the loss function in Equation 1 is effective in transferring clean performance, it does not seem to be able to transfer robustness against adversarial examples (Goldblum et al., 2020) defined as

$$\boldsymbol{x}' = \underset{\|\boldsymbol{x}' - \boldsymbol{x}\| \leq \varepsilon}{\arg\max} \; \text{CE}(f(\boldsymbol{x}'), y),$$

which can be generally obtained using Fast Gradient Sign Method (FGSM) (Goodfellow et al., 2014) or Projected Gradient Descent (PGD) (Madry et al., 2018).

For this reason, recent works have proposed different *robust knowledge distillation* (RKD) alternatives to address this problem (Goldblum et al., 2020; Zi et al., 2021; Zhu et al., 2021; Shao et al., 2021). Specifically, Adversarially Robust Distillation (ARD) (Goldblum et al., 2020) proposes to add a regularization term that encourages robustness by matching $f_S$ and $f_T$ at the output of adversarial and clean examples, i.e.,

$$\text{ARD}(\boldsymbol{x}, y) = (1 - \lambda)\text{CE}(f_S(\boldsymbol{x}), y) + \lambda\text{KL}(f_S(\boldsymbol{x}'), f_T(\boldsymbol{x})). \tag{2}$$

where $f_S(\boldsymbol{x}')$ is the probability output of the student model on the adversarial example $\boldsymbol{x}'$. Meanwhile, Robust Soft Label Adversarial Distillation (RSLAD) (Zi et al., 2021) avoids the use of clean labels and focuses only on matching the models' outputs, i.e.,

$$\text{RSLAD}(\boldsymbol{x}) = (1 - \lambda)\text{KL}(f_S(\boldsymbol{x}), f_T(\boldsymbol{x})) + \lambda\text{KL}(f_S(\boldsymbol{x}'), f_T(\boldsymbol{x})). \tag{3}$$

Compared with the ARD method, this slight modification helps the student model to achieve better results, so that RSLAD is currently the most promising RKD method to date.

Yet, the above losses share a potential issue: they do not encourage matching the outputs of the teacher and the student in the adversarial regions. However, Beyer et al. (2022) showed that, for CKD, the most effective distillation protocols are those that match the teacher's and the student's outputs for all input points. They observe better accuracy when function matching is applied to data augmentation, but it is unclear if this benefit also occurs when applied on adversarial regions or the effect it has on robustness. To better understand the effect of function matching in RKD, we, thus, introduce an additional RKD loss, Adversarial Knowledge Distillation (AKD), which matches the outputs in the adversarial regions, i.e.,

$$\text{AKD}(\boldsymbol{x}, y) = \text{CE}(f_S(\boldsymbol{x}'), f_T(\boldsymbol{x}')). \tag{4}$$

where, inspired by adversarial training (Madry et al., 2018), we minimize the cross-entropy loss on the adversarial example $\boldsymbol{x}'$ against the corresponding teacher prediction $f_T(\boldsymbol{x}')$.

In what follows, we will compare the performance of these loss functions in different settings with the goal of understanding when knowledge distillation can be used effectively to increase the student model robustness.

## 3 Ablation studies

We now present our general ablation studies in which we motivate and analyze different parameter and optimization choices and explore their impact in the context of RKD. To that end, we will compare the clean and robust performance obtained by optimizing the RSLAD, ARD and AKD losses depending on these choices.

We will only consider the case where the student and teacher models share the same architecture. Specifically, we will mostly use a ResNet-18 in the majority of our experiments[1]. The training details of all our experiments can be found in Appendix A. However, as a general rule, we train our teachers using stochastic gradient descent (SGD) with an exponential learning rate decay for 100 epochs, unless early-stopped. Afterwards, we train the students using the ARD, RSLAD and AKD loss functions in Equation 2, Equation 3 and Equation 4, respectively, using SGD with a cyclic learning rate decay for 100 epochs. We find this number to be sufficient on CIFAR10 when no additional synthetic data is incorporated. In both cases, the initial learning rate is 0.1 and the batch size is 256. Unless stated otherwise, we use PGD with 7 iterative steps (PGD-7) (Madry et al., 2018) and AutoAttack (Croce & Hein, 2020) $l_\infty$ perturbations of size $\varepsilon = 8/255$ to perform adversarial training and evaluate robustness, respectively. We use PGD-7 for faster training without risk of catastrophic overfitting (Andriushchenko & Flammarion, 2020), and the more costly AutoAttack to ensure we are measuring robustness correctly.

Our goal with the following experiments is thus to emphasize the situations under which we can use RKD to push the Pareto frontier between clean accuracy and adversarial robustness. That is, we want to find models that dominate any combination of clean and adversarial accuracy. Particularly, we will study the effect of model capacity, teacher robustness, early-stopping, label mixing, number of iterations of the training attacks and the use of ensembles of multiple teachers.

### 3.1 Label mixing

The original motivation of RKD was to transfer robustness from big models to smaller architectures, but our main goal is to surpass the performance of the robust teacher. The success of self-distillation (Dong et al., 2019) has shown that, sometimes, mixing a teacher's outputs with the original labels yields better results than training with the clean labels alone. We therefore explore the use of label mixing in RKD.

To do so, we use a slight generalization of ARD, RSLAD and AKD, denoted with a plus sign, in which we mix the original labels alongside the teacher outputs. This results in the following loss functions:

$$\text{ARD}_+(\boldsymbol{x}, y) = (1 - \lambda)\text{CE}(f_S(\boldsymbol{x}), y) + \lambda\text{KL}(f_S(\boldsymbol{x}'), \alpha f_T(\boldsymbol{x}) + (1 - \alpha)y), \tag{5}$$

$$\text{RSLAD}_+(\boldsymbol{x}, y) = (1 - \lambda)\text{KL}(f_S(\boldsymbol{x}), \alpha f_T(\boldsymbol{x}) + (1 - \alpha)y) + \lambda\text{KL}(f_S(\boldsymbol{x}'), \alpha f_T(\boldsymbol{x}) + (1 - \alpha)y), \tag{6}$$

$$\text{AKD}_+(\boldsymbol{x}, y) = \text{CE}(f_S(\boldsymbol{x}'), \alpha f_T(\boldsymbol{x}') + (1 - \alpha)y), \tag{7}$$

where $f_T(\boldsymbol{x})$ is the probability output of the teacher model on the data point $\boldsymbol{x}$, and $\alpha \in [0, 1]$ controls the mixing of the distilled labels and the original ones.

Label mixing allows the student model to compensate for the variable quality of the distilled labels during training. As we will see, this is an important feature in the context of adversarial training where the existence of hard-to-learn samples and label noise can severely hamper the final performance of a model (Liu et al., 2021; Dong et al., 2021). Furthermore, we note that the label mixing used in Equation 5 and 6 penalizes model over-confidence and changes the optimum of the optimization to $\alpha f_T(\boldsymbol{x}) + (1 - \alpha)y$. This can help reduce the model Lipschitz constant around the data points (i.e., maximum gradient with respect to the input), thus making the student more robust (Cisse et al., 2017). Moreover, training only on the clean labels can also be detrimental: label noise significantly hurts the performance of adversarially trained models (Sanyal et al., 2020), while standardly trained models tend to memorize certain parts of the training set (Arpit et al., 2017; Rahaman et al., 2019).

We compare in Figure 1 the performance of the student model when we change the value of $\alpha$. Note that increasing/decreasing $\alpha$ is a way to control our confidence on the teacher outputs, and thus to control the impact of KD. We observe that label mixing is especially helpful when we use a standardly trained teacher, as it avoids transferring the teacher susceptibility to adversarial perturbations (at $\alpha = 1$ the student has trivial robust accuracy) while transferring some of its clean performance. For adversarially trained teachers, we observe that, like early stopping, label mixing is necessary to maximize the student robustness. While the teacher outputs transfer robustness effectively, slightly mixing the original labels can help correcting

---

[1]We obtain similar results for different model architectures (see Figure 4 in the Appendix)

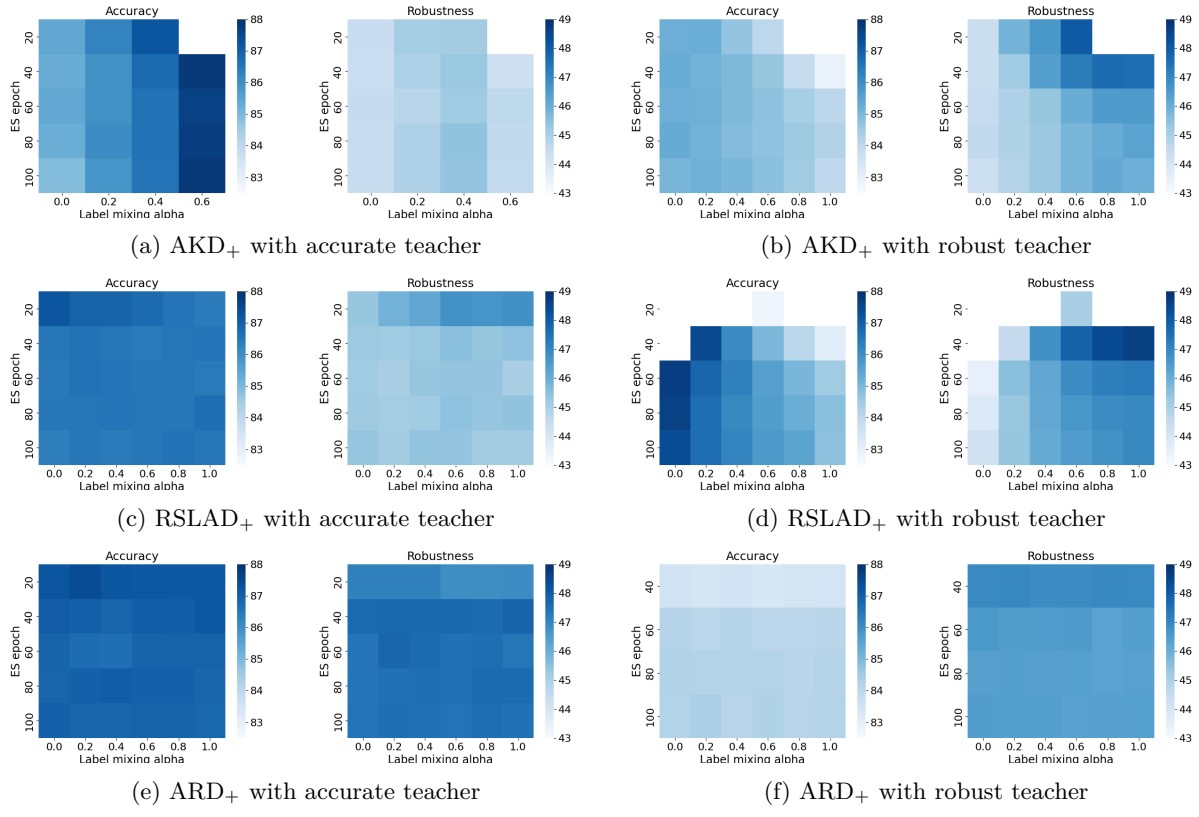

Figure 1: Effect of early stopping and label mixing on the AKD$_+$, RSLAD$_+$ and ARD$_+$ methods. We use the ResNet-18 architecture for the teacher and the student, on the CIFAR10 dataset (Krizhevsky & Hinton, 2009). We removed the combinations that result in either very low accuracy or very low robustness.

the samples that the robust teacher struggles to learn (Liu et al., 2021; Dong et al., 2021). In practice, we found that $\alpha$ values around 0.3 and 0.7 are most beneficial for student models trained with standardly and adversarially trained teachers, respectively. Similar results can be obtained for teachers of a higher capacity, as we will see in the next section.

## 3.2 Early stopping

We now study the use of early stopping during the teacher training to increase the robustness of the student model, while also reducing the computational cost of training the teacher model in RKD. In fact, some CKD methods use early-stopped teachers to improve the accuracy of the student model, but it is only effective on situations where there is a mismatch in the capacity of the teacher and student models (Cho & Hariharan, 2019; Yang et al., 2018). We show this is not the case for RKD where early stopping is generally beneficial.

To illustrate the benefits of early stopping in increasing robustness of models of similar capacity, we show in Figure 1 how the performance of the student model varies depending on the early-stopping point in the training of its teacher. We can observe there is an optimal early stopping point for adversarially trained teachers where the student robustness is maximized at the cost of some clean accuracy decrease.

This increase in student model robustness is due to multiple compounded effects. First, we note that in the later stages of adversarial training, the teacher network is memorizing rather than learning new features (Arpit et al., 2017; Rahaman et al., 2019). This may hinder the transfer of robustness to the student network (Liu et al., 2021; Dong et al., 2021). Second, early-stopping the teacher allows to distill from a model which is less confident of its outputs thus maximizing the potential of learning dark knowledge, like implicit similarity between classes (Mirzadeh et al., 2020). And finally, using an early-stopped adversarially trained

Table 2: Best performance of the RKD methods (i.e., $ARD_+$, $RSLAD_+$ and $AKD_+$) against fine-tuning the teacher for the same number of epochs, on the CIFAR10 dataset. The analysis is performed on ResNet-18 models where the number of channels in each layer is scaled by different values ("layer size" column). Hyperparameter choices for RKD-Acc. and RKD-Rob. aim to improve accuracy or robustness without reducing the baseline robustness and accuracy by more than 2%, respectively. We use AutoAttack (Croce & Hein, 2020) $l_\infty$ adversarial perturbations of size $\varepsilon = 8/255$ to test the model robustness. ES: early-stopping epoch, $\alpha$: label mixing parameter

| Layer size | Method | Clean | Robust | Best loss | Hyperparameters |
|---|---|---|---|---|---|
| *1 | Only teacher | $85.54_{\pm.18}$ | $46.33_{\pm.05}$ | | |
| | RKD-Acc. | $\mathbf{87.40}_{\pm.26}$ | $47.16_{\pm.14}$ | $ARD_+$ | ES=20, $\alpha$=0.2 |
| | RKD-Rob. | $84.10_{\pm.37}$ | $\mathbf{48.40}_{\pm.25}$ | $RSLAD_+$ | ES=40, $\alpha$=0.8 |
| *1/4 | Only teacher | $78.54_{\pm.16}$ | $41.78_{\pm.34}$ | | |
| | RKD-Acc. | $\mathbf{80.26}_{\pm.10}$ | $41.39_{\pm.27}$ | $AKD_+$ | ES=40, $\alpha$=0.5 |
| | RKD-Rob. | $77.15_{\pm.28}$ | $\mathbf{42.73}_{\pm.23}$ | $AKD_+$ | ES=100, $\alpha$=0.8 |
| *1/16 | Only teacher | $61.92_{\pm.49}$ | $\mathbf{31.93}_{\pm.25}$ | | |
| | RKD-Acc. | $\mathbf{64.82}_{\pm.25}$ | $30.16_{\pm.22}$ | $AKD_+$ | ES=80, $\alpha$=0.5 |
| | RKD-Rob. | $60.59_{\pm.18}$ | $\mathbf{32.14}_{\pm.27}$ | $AKD_+$ | ES=80, $\alpha$=0.2 |

teacher can prevent negative effects stemming from catastrophic or robust overfitting (Andriushchenko & Flammarion, 2020; Rice et al., 2020). This final benefit can greatly reduce the computational cost of adversarially training the teacher, as it enables the use of single-iteration adversarial attacks during training, as we will see in Subsection 3.5. When considering the potential cost reduction achieved through early-stopping, the comparison made previously in Table 2 becomes even more advantageous for RKD methods.

## 3.3 Model capacity

For this experiment[2], we use a ResNet-18 (He et al., 2016) architecture where the number of channels in every layer is resized by a scalar quantity, on the CIFAR10 dataset (Krizhevsky & Hinton, 2009). This way, we can study the influence of the capacity of the network on the gain in performance obtained from knowledge distillation, compared to longer training. For the learning rate schedules, we have considered cyclic, step and exponential decay. We compare training the teacher for 200 epochs versus using knowledge distillation, where both the teacher and the student are trained for 100 epochs. Since both teacher and student networks have the same capacity, both methodologies have equivalent computational cost.

We provide the results obtained in this experiment in Table 2. We see that, while RKD methods can always be effective in improving the clean performance of the student, they are only effective in transferring robustness when the models have high enough capacity. When scaling the ResNet-18 network layers down 16 times, we observe that it is better to adversarially train the single model for longer, rather than to perform RKD with the purpose of increasing the model robustness. We believe the limited capacity has a negative effect in the effectiveness of RKD methods, which rely on the teacher giving reliable outputs and the student being able to learn from these more nuanced representations. The increase in the optimal early-stopping epoch for lower capacity models supports teacher underfitting as the main cause of RKD low performance. Based on these results, RKD should be used over longer training to improve the robustness of models of similar or bigger size than the ResNet-18 architecture. We will now provide guidelines to make RKD methods as effective as possible in terms of the student model accuracy and robustness.

## 3.4 Teacher robustness

We now study the impact of the robustness of the teacher on the student performance. Intuitively, this should be the most important factor, since we partly distill the teacher outputs to train the student model.

---

[2]The code will be open-sourced upon acceptance.

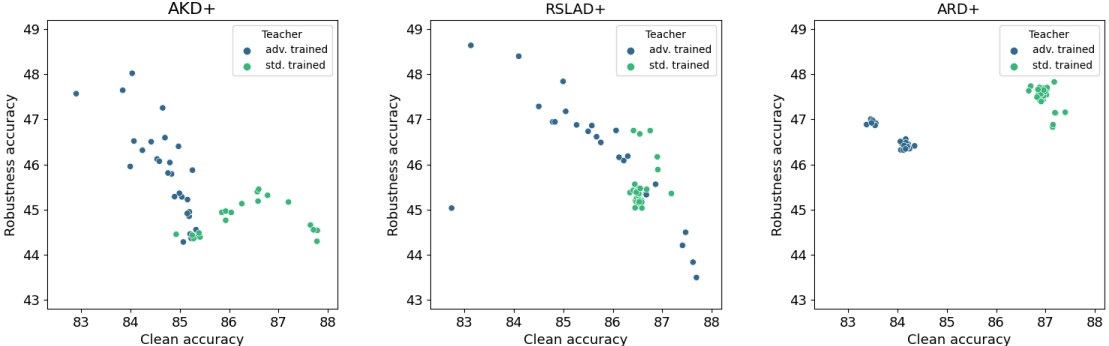

Figure 2: Effect of teacher robustness on the $AKD_+$, $RSLAD_+$ and $ARD_+$ methods. Each point represents the performance of the student model for different label mixing and early-stopping values. We use the ResNet-18 architecture for the teacher and the student, on the CIFAR10 dataset (Krizhevsky & Hinton, 2009).

Table 3: Impact of differently trained teachers on the student performance on the $AKD_+$ and $RSLAD_+$ methods, on the CIFAR10 dataset. We use AutoAttack (Croce & Hein, 2020) $l_\infty$ adversarial perturbations of size $\varepsilon = 8/255$ to test the model robustness. For comparison, the performance of an adversarially trained teacher is $85.54_{\pm.18}$ and $46.33_{\pm.05}$ of clean and robust accuracy, respectively.

| Teacher training | $AKD_+$ | | $RSLAD_+$ | |
| --- | --- | --- | --- | --- |
| | Clean | Robust | Clean | Robust |
| ST | $\mathbf{87.21}_{\pm.03}$ | $45.17_{\pm.63}$ | $\mathbf{87.19}_{\pm.30}$ | $45.36_{\pm.33}$ |
| $AT_{FFGSM}$ | $85.18_{\pm.25}$ | $47.51_{\pm.17}$ | $85.20_{\pm.27}$ | $47.59_{\pm.23}$ |
| $AT_{PGD-7}$ | $84.02_{\pm.19}$ | $\mathbf{48.08}_{\pm.25}$ | $83.13_{\pm.61}$ | $\mathbf{48.64}_{\pm.34}$ |

Thus, if the teacher is not robust enough, matching its outputs by distillation will result in the student model being weakly robust as well.

In Figure 2, we compare the results of the student model when the teacher is trained using standard or adversarial training respectively. For $AKD_+$, we consistently observe that standardly trained teachers tend to boost the clean performance of the student while preserving robustness, while adversarially trained teachers tend to increase adversarial performance, at the cost of some decrease in clean accuracy. For $RSLAD_+$, we need to use an adversarially trained teacher to maximize either student robustness or accuracy, while using standardly trained teachers generates more balanced results. We find that $AKD_+$ is the best method when maximing one of the metrics, while $RSLAD_+$ can be preferable for balanced performance. For $ARD_+$, we find that only standardly trained teachers should be used, as this maximizes the performance gained by the student model, both in terms of accuracy and adversarial robustness.

### 3.5 Adversarial training of the teacher

When using an adversarially trained teacher, there are some particular design choices to make. In particular, we study the effect of the adversarial attack used in training, and the strength of the attack given by the size of the $l_\infty$ ball that is considered.

In Table 3, we distinguish between training the teacher using PGD with 7 iterations (PGD-7) (Madry et al., 2018) and Fast-FGSM (FFGSM) (Wong et al., 2020) adversarial attacks. The latter permits to obtain slightly less robust models in a fraction of the time thanks to only requiring one gradient step to compute the adversarial example. In these experiments, we optimize the early stopping epoch of the teacher, and the label mixing parameter, to maximize robust performance when the teacher is trained adversarially, and clean performance otherwise. We observe that, even if they do not boost the robust performance as much as the PGD-7 models, the weaker FFGSM models tend to achieve better trade-offs between robustness and accuracy.

Table 4: Effect of using different $l_\infty$ ball constraints to adversarially train the teacher model on the $AKD_+$ and $RSLAD_+$ methods. We use the ResNet-18 architecture for both teacher and student models, on the CIFAR10 dataset. We use AutoAttack (Croce & Hein, 2020) $l_\infty$ adversarial perturbations of size $\varepsilon = 8/255$ to test the model robustness.

| $l_\infty$ ball size ($\varepsilon$) | $AKD_+$ | | $RSLAD_+$ | |
| --- | --- | --- | --- | --- |
| | Clean | Robust | Clean | Robust |
| 0 | $\mathbf{87.21}_{\pm.03}$ | $45.17_{\pm.63}$ | $\mathbf{87.19}_{\pm.30}$ | $45.36_{\pm.33}$ |
| 2/255 | $\mathbf{87.37}_{\pm.56}$ | $46.71_{\pm.17}$ | $\mathbf{87.57}_{\pm.13}$ | $46.68_{\pm.24}$ |
| 8/255 | $84.02_{\pm.19}$ | $\mathbf{48.08}_{\pm.25}$ | $84.10_{\pm.37}$ | $\mathbf{48.40}_{\pm.25}$ |

Finally, inspired by works from the adversarial robustness literature that claim that adversarial training with very small perturbations can improve generalization (Xie et al., 2020; Bochkovskiy et al., 2020), we also look at the effect of the adversarial perturbation region size in RKD. By varying the $\varepsilon$ value for the strength of the attack used to train the teacher model adversarially (see Table 4), we observe that a small $\varepsilon$ value results in higher accuracy and robustness compared to standard training. This increase in generalization can be transferred to the student, even when the student is still trained using perturbations with the standard $\varepsilon$ value of 8/255 used in CIFAR datasets.

### 3.6 Ensemble of teachers

Until this point, we have only trained a single teacher. However, we now show that using an ensemble of similarly trained teachers, which only differ on their random initialization, can distill robustness even further. This technique is commonly used in CKD (Chebotar & Waters, 2016; Cui et al., 2017; Freitag et al., 2017) to improve clean accuracy, but it is unclear if it should work to increase robustness as well. Especially, since an ensemble of models is no more robust than each of its individual models (see Table 12 in the Appendix).

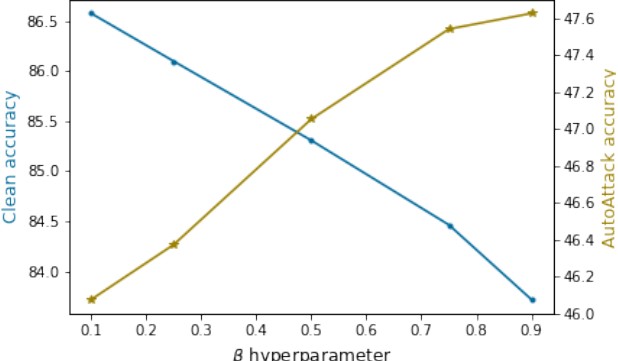

Figure 3: Effect of varying the mixing of two standardly trained and adversarially trained teachers using $AKD_+$. We use the ResNet-18 architecture for both the teacher and student models, on the CIFAR10 dataset.

Table 5 shows an equivalent analysis to Subsection 3.4 for the $AKD_+$ loss function[3], but with an ensemble of similarly trained teachers rather than a single one. To that end, we can generalize the $AKD_+$ objective for an ensemble of teachers $\{f_{T_i}\}_{i=1}^M$ as

$$\text{AKD}_+(\boldsymbol{x}, y) = \text{CE}\left(f_S(\boldsymbol{x}'),\ \alpha \sum_{i=1}^M \beta_i f_{T_i}(\boldsymbol{x}') + (1-\alpha)y\right)$$

where $M$ is the number of networks in the ensemble and $\beta_i$ controls the mixing of all the individual teachers ($\beta_i \geq 0$, $\sum_i \beta_i = 1$). For our analysis, we use $\beta_i = 1/M$.

Using an ensemble of teachers improves both clean accuracy and robust performance across most training configurations. It notably enhances accuracy when the teachers are standardly trained and further improves robustness when the teachers are adversarially trained. Based on the results, it is better to train an ensemble of four models from scratch with FFGSM adversarial training than a single model with PGD-7, which has a similar computational cost[4]. Moreover, the cost of training an ensemble of standardly trained models for the teacher may be negligible compared to the inherent cost of adversarially training the student.

---

[3]The results for the $RSLAD_+$ loss function can be found in Subsection B.3

[4]For computational cost comparison, $AT_{FFGSM}$ is assumed to cost 2 times more than standard training, and $AT_{PGD-7}$ is assumed to cost 8 times more

Table 5: Effect of using an ensemble for the teacher model on the $AKD_+$ method. We use the ResNet-18 architecture for both student and teacher models. We use AutoAttack (Croce & Hein, 2020) $l_\infty$ adversarial perturbations of size $\varepsilon = 8/255$ to test the model robustness.

| Dataset | Teacher training | 1 model | | 4 models | |
|---|---|---|---|---|---|
| | | Clean | Robust | Clean | Robust |
| CIFAR10 | ST | $\mathbf{87.21}_{\pm.03}$ | $45.17_{\pm.63}$ | $\mathbf{87.79}_{\pm.10}$ | $45.58_{\pm.17}$ |
| | $AT_{FFGSM}$ | $85.18_{\pm.25}$ | $47.51_{\pm.17}$ | $84.11_{\pm.10}$ | $\mathbf{49.15}_{\pm.22}$ |
| | $AT_{PGD\text{-}7}$ | $84.02_{\pm.19}$ | $\mathbf{48.08}_{\pm.25}$ | $83.03_{\pm.18}$ | $\mathbf{49.33}_{\pm.08}$ |
| CIFAR100 | ST | $\mathbf{62.73}_{\pm.59}$ | $21.44_{\pm.36}$ | $\mathbf{63.52}_{\pm.57}$ | $21.07_{\pm.25}$ |
| | $AT_{FFGSM}$ | $58.41_{\pm.09}$ | $\mathbf{24.53}_{\pm.11}$ | $59.33_{\pm.12}$ | $\mathbf{25.36}_{\pm.11}$ |
| | $AT_{PGD\text{-}7}$ | $58.96_{\pm.97}$ | $23.77_{\pm.68}$ | $57.93_{\pm.02}$ | $\mathbf{25.34}_{\pm.01}$ |

Table 6: Performance comparison of the teacher trained using state-of-the-art techniques (AT+) and the student trained using $AKD_+$, $RSLAD_+$ or $ARD_+$, when 1 million data samples generated using DDPM are added to the training set (Rebuffi et al., 2021). The student and the teacher use the WideResNet-28 (Zagoruyko & Komodakis, 2016b) with swish activations (Hendrycks & Gimpel, 2016) architecture. We use AutoAttack (Croce & Hein, 2020) $l_\infty$ adversarial perturbations of size $\varepsilon = 8/255$ to test the model robustness.

| Dataset | Training type | $\alpha$ | ES | Clean | PGD-20 | Robust |
|---|---|---|---|---|---|---|
| CIFAR10 | AT+ | — | No | $87.95_{\pm.18}$ | $61.11_{\pm.40}$ | $58.76_{\pm.72}$ |
| | AKD | 1.0 | No | $88.25_{\pm.22}$ | $62.55_{\pm.04}$ | $60.37_{\pm.25}$ |
| | AKD | 1.0 | ep. 100 | $86.75_{\pm.19}$ | $60.27_{\pm.11}$ | $58.55_{\pm.13}$ |
| | $AKD_+$ | 0.8 | No | $88.35_{\pm.14}$ | $\mathbf{62.96}_{\pm.36}$ | $\mathbf{60.78}_{\pm.43}$ |
| | RSLAD | 1.0 | No | $88.07_{\pm.16}$ | $62.15_{\pm.04}$ | $\mathbf{60.55}_{\pm.14}$ |
| | $RSLAD_+$ | 0.8 | No | $\mathbf{88.67}_{\pm.03}$ | $62.09_{\pm.05}$ | $\mathbf{60.65}_{\pm.01}$ |
| | ARD | 1.0 | No | $88.23_{\pm.04}$ | $62.12_{\pm.35}$ | $59.89_{\pm.30}$ |
| | $ARD_+$ | 0.8 | No | $88.03_{\pm.00}$ | $61.94_{\pm.06}$ | $59.53_{\pm.62}$ |
| CIFAR100 | AT+ | — | No | $61.96_{\pm.23}$ | $31.99_{\pm.25}$ | $29.37_{\pm.35}$ |
| | $AKD_+$ | 0.8 | No | $\mathbf{62.13}_{\pm.09}$ | $\mathbf{33.13}_{\pm.32}$ | $\mathbf{30.48}_{\pm.50}$ |

Finally, one of the main advantages of using an ensemble of teacher models is that we are not constrained to use one type of training for all models. This allows us to have much more control when choosing the trade-off between accuracy and robustness, since we can use different $\beta_i$ values for each type of model. We show in Figure 3 the performance of the student when mixing two standardly trained and adversarially trained teacher models, where $\beta$ determines the mixing weight of the adversarially trained teacher model. We observe that the change in performance as we increase $\beta$ is quite smooth. This permits to obtain student models with more desirable clean and robust performance trade-offs.

Overall, we have presented different ways to train a teacher model and distill its knowledge using RKD and found that using early stopping, label mixing and an ensemble of teachers results in better overall performance. Particularly, we found that student accuracy and robustness can be maximized by using small-$\varepsilon$ and typical-$\varepsilon$ adversarial training, respectively. And finally, we have shown that the teacher can be trained with weak but much cheaper attacks with no drawback in performance. In the next Section, we will thus apply all these insights with the aim to further push the robustness of state-of-the-art models.

## 4 Implementing our guidelines

To clearly demonstrate that RKD can serve as an effective plugin method to enhance the performance of naive adversarially trained models, we extend our guidelines to more advanced robust models. Specifically, we combine RKD with state-of-the-art adversarial training techniques (Rebuffi et al., 2021; Wang et al., 2023).

Table 7: Performance comparison of the teacher trained using state-of-the-art techniques (AT+) and the student trained using $AKD_+$ or $RSLAD_+$, when 1 million data samples generated using EDM are added to the training set (Karras et al., 2022). The student and the teacher use the WideResNet-28 (Zagoruyko & Komodakis, 2016b) with swish activations (Hendrycks & Gimpel, 2016) architecture. We use AutoAttack (Croce & Hein, 2020) $l_\infty$ adversarial perturbations of size $\varepsilon = 8/255$ to test the model robustness.

| Dataset | Training type | $\alpha$ | ES | Clean | PGD-20 | Robust |
|---|---|---|---|---|---|---|
| CIFAR10 | AT+ | — | No | $\mathbf{92.84}_{\pm.04}$ | $64.12_{\pm.11}$ | $61.69_{\pm.15}$ |
| | AKD | 1.0 | No | $92.48_{\pm.14}$ | $63.98_{\pm.12}$ | $61.79_{\pm.04}$ |
| | $AKD_+$ | 0.8 | No | $92.67_{\pm.07}$ | $\mathbf{64.51}_{\pm.21}$ | $\mathbf{62.08}_{\pm.01}$ |
| | $RSLAD_+$ | 1.0 | No | $92.51_{\pm.06}$ | $63.51_{\pm.15}$ | $61.78_{\pm.16}$ |
| | $RSLAD_+$ | 0.8 | No | $\mathbf{92.76}_{\pm.09}$ | $63.62_{\pm.11}$ | $61.85_{\pm.16}$ |
| CIFAR100 | AT+ | — | No | $\mathbf{71.53}_{\pm.24}$ | $35.58_{\pm.45}$ | $32.60_{\pm.30}$ |
| | AKD | 1.0 | No | $70.77_{\pm.36}$ | $\mathbf{36.91}_{\pm.23}$ | $\mathbf{34.31}_{\pm.47}$ |
| | $AKD_+$ | 0.8 | No | $70.84_{\pm.07}$ | $\mathbf{37.05}_{\pm.40}$ | $\mathbf{34.35}_{\pm.18}$ |
| | $RSLAD_+$ | 1.0 | No | $70.54_{\pm.16}$ | $36.56_{\pm.13}$ | $33.98_{\pm.04}$ |
| | $RSLAD_+$ | 0.8 | No | $70.80_{\pm.23}$ | $36.47_{\pm.16}$ | $33.78_{\pm.07}$ |

Table 8: Effect of using ensembles of teachers for a student trained using $AKD_+$, when 1 million data samples generated using DDPM are added to the training set (Rebuffi et al., 2021). The student and the teacher use the WideResNet-28 (He et al., 2016; Zagoruyko & Komodakis, 2016b) with swish activations (Hendrycks & Gimpel, 2016) architecture. We use AutoAttack (Croce & Hein, 2020) $l_\infty$ adversarial perturbations of size $\varepsilon = 8/255$ to test the model robustness.

| Dataset | Training type | $\alpha$ | ES | Clean | PGD-20 | Robust |
|---|---|---|---|---|---|---|
| CIFAR10 | AKD (1 teacher) | 1.0 | No | 88.25 | 62.55 | 60.37 |
| | $AKD_+$ (1 teacher) | 0.8 | No | **88.35** | **62.96** | **60.78** |
| | AKD (4 teachers) | 1.0 | No | 87.99 | 62.53 | 60.55 |
| | $AKD_+$ (4 teachers) | 0.8 | No | **88.41** | **63.02** | **61.07** |

To enhance model robustness, the authors in (Rebuffi et al., 2021) employ various strategies, including larger architectures (e.g., WideResNet-28 (He et al., 2016; Zagoruyko & Komodakis, 2016b) with swish activations (Hendrycks & Gimpel, 2016)), model weight averaging (Izmailov et al., 2018), additional data from Denoising Diffusion Probabilistic Models (DDPM)(Ho et al., 2020), and data augmentation techniques like Cutout(DeVries & Taylor, 2017). We use these techniques, following the default hyperparameters in (Rebuffi et al., 2021), for both teacher and student training. Notably, RKD methods can be easily combined with these enhancements. The ARD+ method was not used in this experiment, as it does not benefit from the proposed early-stopping and label mixing techniques outlined in this study and the $AKD_+$ and RSLAD+ methods demonstrate superior results.

Based on our previous experiments, we propose to use a high value of $\alpha$ for the label mixing (e.g. 0.8), since we have confidence on the quality of the output given by a state-of-the-art robust teacher. When considering early-stopping, the inclusion of synthetic data can alter the expected outcomes, as it has a comparable effect in preventing model memorization. Thus, we find early-stopping to only be significant on smaller sized datasets. Finally, we expect FFGSM to not be viable without early-stopping due to catastrophic overfitting. Thus, we advice to only consider using an ensemble of teachers trained with PGD when the computational cost is not a limitation, as it can give a slight edge compared to only using one teacher (Table 8).

In Table 6, we show empirically the effects of using label mixing and early stopping. We find that $AKD_+$ and RSLAD+ significantly enhance student performance compared to the teacher, even without using label mixing. Interestingly, $AKD_+$ slightly outperforms RSLAD, which may be due to the matching of logits from adversarial examples in $AKD_+$, compensating for the teacher's confidence from training on larger data quantities. We evaluate the impact of our design choices, finding that early stopping is less effective due

to data augmentation preventing memorization. However, label mixing further improves clean and robust performance compared to using only distilled labels, consistent with our previous results.

Extending our analysis to the current state-of-the-art method in robustness (Wang et al., 2023), where Elucidating Diffusion Models (EDM)(Karras et al., 2022) replace DDPM for synthetic data generation, we observe variations in the effectiveness of RKD models. When DDPM is used, substantial performance improvements are observed. However, with advanced diffusion models like EDM, the student's performance on CIFAR10 closely aligns with that of the teacher (Table 7). Surprisingly, for the more challenging CIFAR100 dataset (Krizhevsky & Hinton, 2009), RKD achieves better robustness at the cost of some accuracy decrease. We believe this is a result of the hierarchical class structure of the CIFAR100 dataset, which results in many classes being very similar to others. This information can be transferred efficiently to the student using KD, since the distilled probabilities of similar classes will tend to be more correlated (Hinton et al., 2015).

The discrepancy in performance can be attributed to the characteristics and properties inherent in different diffusion models used for synthetic data generation. We speculate that EDM, being a superior diffusion model based on Fréchet Inception Distance (FID), produces synthetic data that closely resembles the complexity and distribution of real-world data. As model robustness relies on accurate labeling, improved data quality reduces the impact of knowledge distillation in correcting labels. Consequently, a substantial amount of data generated by EDM may already capture much of the knowledge transferable from a teacher model with the same architecture as the student, resulting in more similar performance levels.

## 5 Related work

While RKD methods (Goldblum et al., 2020; Zi et al., 2021; Shao et al., 2021) mainly focus on compressing robust models, recent studies have shown that knowledge distillation can enhance model performance (Nix et al., 2023; Wu et al., 2023; Mandal & Gao, 2023). In our work, we incorporate unexplored design choices from CKD to our proposed $AKD_+$, $RSLAD_+$ and $ARD_+$ loss functions. Label mixing is inspired from self-distillation (Dong et al., 2019), in which the outputs of the teacher are mixed with the clean labels. Moreover, there are other CKD works that proposed to use an ensemble of teachers to improve clean performance (Chebotar & Waters, 2016; Cui et al., 2017; Freitag et al., 2017), which we show can also increase robustness when used on RKD methods.

Our $AKD_+$ loss function also contrasts with other CKD (Hinton et al., 2015) and RKD methods in that function matching happens implicitly in the optimization instead of as an additional regularization term. Particularly, we differ from RKD methods in that we use function matching exclusively in the adversarial region, inspired by (Beyer et al., 2022; Dong et al., 2019). The increased improvement of our method on the more complex CIFAR100 dataset is consistent with the claim that KD encodes the semantic similarity between classes (Hinton et al., 2015; Furlanello et al., 2018; Yang et al., 2018). Moreover, the fact that we obtain better performance using early stopping has been theorized in the context of memorization, where it is claimed that neural networks learn some of the most important features very early and rely on memorization in the later steps of the training (Arpit et al., 2017; Dong et al., 2019).

In the context of adversarial robustness, we find that $AKD_+$ penalizes overconfidence on easier examples and improves performance on difficult-to-learn regions (see Appendix). This aligns with recent adversarial literature indicating the difficulty of achieving robustness due to network limitations in memorizing harder instances or susceptibility to label noise (Sanyal et al., 2020; Liu et al., 2021; Dong et al., 2021; Rahaman et al., 2019). It is also known that networks with reduced Lipschitz constants exhibit greater robustness (Cisse et al., 2017). In this work, we show that $AKD_+$ can complement the current state-of-the-art methodology, which mainly combines adversarial training (Madry et al., 2018; Zhang et al., 2019) with data generation using diffusion models (Ho et al., 2020; Karras et al., 2022).

## 6 Conclusion

In this work, we presented a thorough analysis showing how RKD methods can be used to boost the model robustness, surpassing state-of-the-art performance. We gave guidelines to use early stopping, label

mixing, ensembling, and small-epsilon adversarial training to boost the student model performance, and show accordingly how new robust KD algorithms: $AKD_+$, $RSLAD_+$ and $ARD_+$, achieve higher robustness than their equivalent RKD methods.

In the future, we plan to develop understanding of the underlying properties of the labels generated by label mixing, which make them superior to both the teacher outputs and the original labels in terms of increasing the student model robustness. This could be a key insight to further refine the data used to train the model, compared to what can be actually achieved with knowledge distillation.

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

# A Details on the experimental setup

## A.1 Architectures and training

For all our experiments in Section 3 and Section C.2, we use the ResNet-18 architecture He et al. (2016) for both teacher and student models. For Section 4, we use the WideResNet-28 He et al. (2016); Zagoruyko & Komodakis (2016b) with swish activations Hendrycks & Gimpel (2016) architecture, as proposed in Rebuffi et al. (2021), for both teacher and student models.

We use the same training parameters for both CIFAR10 and CIFAR100 datasets Krizhevsky & Hinton (2009). For both student and teacher models, we use stochastic gradient (SGD) for 100 epochs, with 0.1 as the initial learning rate. Because we found that early-stopping the teacher is beneficial when using AKD, we train it using exponential learning rate decay, decaying at a rate of 0.9 at the end of every epoch. Compared with other schedules, exponential learning rate trains the model more gradually, which helps to tune when the teacher should be early-stopped. For the student, we use the OneCycle learning rate (Smith & Topin, 2019) with two phases and 0.21 as the maximum learning rate, updated every batch. To train the teacher and the student, we use batch size 128 and 256, respectively.

All adversarial examples are crafted with $L_\infty$ perturbations of size $\varepsilon = 8/255$. For all models trained with adversarial training or AKD, we craft the adversarial examples using PGD-7, unless stated otherwise. To evaluate robustness, we use PGD-7 Madry et al. (2018) and AutoAttack Croce & Hein (2020).

# B Extended analysis of our guidelines

## B.1 Trade-offs for compression

We have illustrated that early stopping the teacher and label mixing is beneficial when used to boost the performance of similar capacity models in Sections 3.2 and 3.1. We show that these insights also generalize in the context of compression, where the teacher model is significantly bigger than the student model.

We compare in Figure 4a how the performance of the student model varies depending on when we early-stopped its teacher. For the student, we use the ResNet-18 architecture, while for the teacher we use the WideResNet-28 He et al. (2016); Zagoruyko & Komodakis (2016b) with swish activations Hendrycks & Gimpel (2016) architecture, as proposed in Rebuffi et al. (2021). The student is trained with AKD, using $\alpha = 1.0$ to avoid any confounding effect from label mixing. Meanwhile, the teacher is trained using model weight averaging Izmailov et al. (2018), extra data sampled from generative models Ho et al. (2020) and the default data augmentation techniques DeVries & Taylor (2017) given in Rebuffi et al. (2021). From this plot, we can observe there is an optimal early stopping point where the student robustness is maximized. In contrast, we also see that distilling from an early-stopped teacher reduces the student clean accuracy. These results are consistent with the ones we obtained when using a smaller, adversarially trained, ResNet-18 teacher (see Figure 1).

We compare in Figure 4b the performance of the student model when we change the value of $\alpha$. We consider the teacher has been early-stopped at epoch 100. We observe that the student robustness is maximized with a high $\alpha$ value. This result is consistent with Figure 1, in which the teacher has the same capacity as the student. However, in contrast, we find that the student network robustness is higher, thus showing that AKD can be used for compressing the superior performance of a bigger network, like other RKD methods.

## B.2 Performance on Tiny-ImageNet

Through the paper, we observed that we can use early stopping and label mixing to improve the trade-off between robustness in accuracy in robust knowledge distillation. For this experiment, we apply those techniques to larger-scale datasets. We use the ResNet-18 (He et al., 2016) architecture on the Tiny-Imagenet dataset Le & Yang (2015). We vary the early stopping point and label mixing parameter, and select the results that result in either high accuracy or robustness, without significantly reducing the other metric. We provide the results obtained in this experiment in Table 2.

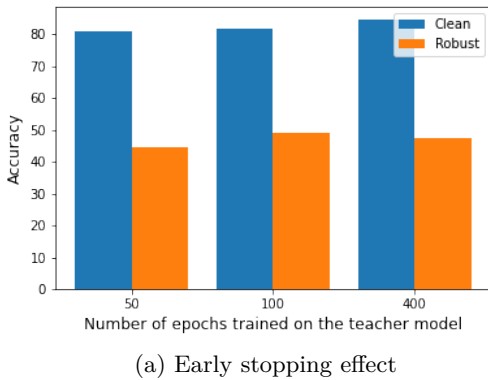

(a) Early stopping effect

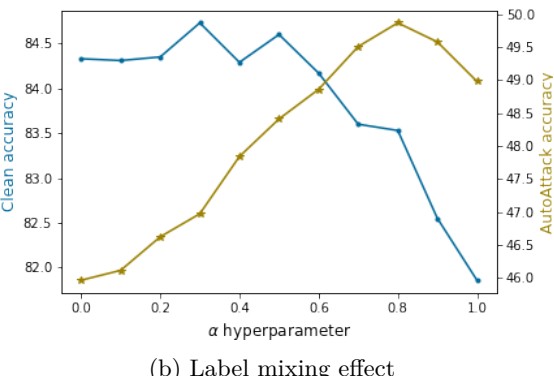

(b) Label mixing effect

Figure 4: Effect of early stopping the teacher (left) and label mixing (right) on the AKD method. We use the state-of-the-art WideResNet-28-10 with swish activations from Rebuffi et al. (2021) as the teacher, and the ResNet-18 for the student, on the CIFAR10 dataset.

Table 9: Best performance of the RKD methods (i.e., ARD$_+$, RSLAD$_+$ and AKD$_+$) against fine-tuning the teacher for the same number of epochs, for the ResNet-18 architecture on the Tiny-Imagenet dataset. Hyperparameter choices for RKD-Acc. and RKD-Rob. aim to improve accuracy or robustness without reducing the baseline robustness and accuracy by more than 2%, respectively. We use AutoAttack $l_\infty$ adversarial perturbations of size $\varepsilon = 8/255$ to test the model robustness. ES: early-stopping epoch, $\alpha$: label mixing parameter

| Method | Clean | Robust | Best loss | Hyperparameters |
|--------|-------|--------|-----------|-----------------|
| Only teacher | $43.96_{\pm.04}$ | $\mathbf{21.77_{\pm.19}}$ | | |
| RKD-Acc. | $\mathbf{46.85_{\pm.40}}$ | $19.92_{\pm.05}$ | AKD$_+$ | ES=30, $\alpha$=0.6 |
| RKD-Rob. | $42.79_{\pm.34}$ | $\mathbf{21.69_{\pm.02}}$ | AKD$_+$ | ES=30, $\alpha$=0.6 |

We see that RKD methods are effective in improving the clean performance of the student. However, due to the lower achievable robustness on Tiny-ImageNet with adversarial training, the improvements in robustness on this dataset are minimal. These results are very similar to the ones obtained in Table 2, when we train low-capacity models on CIFAR10. Moreover, like in that experiment, the best student models had not been early-stopped, which suggests teacher underfitting. This further supports our belief that the limited capacity has a negative effect in the effectiveness of RKD methods, which rely on the teacher giving reliable outputs and the student being able to learn from these more nuanced representations. Thus, we think that for larger and more robust networks, knowledge distillation could be effective on Tiny-ImageNet. The increase in the optimal early-stopping epoch for lower capacity models supports teacher underfitting as the main cause of RKD low performance.

## B.3   Ensemble performance for RSLAD$_+$

In Subsection 3.6, we showed that using an ensemble of teachers can be beneficial when using the AKD loss function. The main reason being that we can use an ensemble of four teachers trained with FFGSM and obtain better results than a single teacher trained with PGD. This is especially interesting when we consider that both options have a similar computational cost.

Here we show this comparison, but for the RSLAD$_+$ loss function. The generalization of this function when using an ensemble of teacher can be described as: $\{f_{T_i}\}_{i=1}^M$ as

$$\text{RSLAD}_+(\boldsymbol{x}, y) = (1-\lambda)\text{KL}(f_S(\boldsymbol{x}), \alpha \sum_{i=1}^M \beta_i f_{T_i}(\boldsymbol{x}) + (1-\alpha)y) + \lambda\text{KL}(f_S(\boldsymbol{x}'), \alpha \sum_{i=1}^M \beta_i f_{T_i}(\boldsymbol{x}) + (1-\alpha)y)$$

Table 10: Effect of using an ensemble for the teacher model on the RSLAD$_+$ method. We use the ResNet-18 architecture for both student and teacher models. We use AutoAttack (Croce & Hein, 2020) $l_\infty$ adversarial perturbations of size $\varepsilon = 8/255$ to test the model robustness.

| Training type | Clean | Robust |
|---|---|---|
| RSLAD$_+$ (1 FFGSM-trained teacher) | $81.91_{\pm.04}$ | $\mathbf{48.27}_{\pm.08}$ |
| RSLAD$_+$ (4 FFGSM-trained teachers) | $82.21_{\pm.30}$ | $\mathbf{48.47}_{\pm.29}$ |
| RSLAD$_+$ (1 PGD-trained teacher) | $\mathbf{84.10}_{\pm.37}$ | $\mathbf{48.40}_{\pm.35}$ |

Table 11: Performance comparison of the teacher trained using state-of-the-art techniques (AT+ or TRADES+) and the student trained using AKD$_+$ or RSLAD$_+$, when 1 million data samples generated using EDM (Karras et al., 2022) are added to the CIFAR10 training set. The student and the teacher use the WideResNet-28 (Zagoruyko & Komodakis, 2016b) with swish activations (Hendrycks & Gimpel, 2016) architecture. We use AutoAttack (Croce & Hein, 2020) $l_\infty$ adversarial perturbations of size $\varepsilon = 8/255$ to test the model robustness.

| Training type | $\alpha$ | ES | AT+ teacher | | TRADES+ teacher | |
|---|---|---|---|---|---|---|
| | | | Clean | PGD-20 | Clean | PGD-20 |
| Use teacher | — | No | 92.84 | 64.12 | 90.89 | 64.36 |
| AKD | 1.0 | No | 92.48 | 63.98 | 90.43 | 63.51 |
| AKD$_+$ | 0.8 | No | 92.67 | 64.51 | 91.59 | 64.43 |
| RSLAD$_+$ | 1.0 | No | 92.51 | 63.51 | 90.34 | 63.91 |
| RSLAD$_+$ | 0.8 | No | 92.76 | 63.62 | 90.49 | 62.94 |

where $M$ is the number of networks in the ensemble and $\beta_i$ controls the mixing of all the individual teachers ($\beta_i \geq 0$, $\sum_i \beta_i = 1$). For our analysis, we use $\beta_i = 1/M$.

In Table 10, we observe an increase in performance when using an ensemble of teachers trained with FFGSM adversarial perturbations. However, this increase is less significant than the increase we obtained with the AKD loss function. We find that for RSLAD$_+$, it is more beneficial to use a single model trained with PGD adversarial perturbations, than an ensemble trained with FFGSM adversarial perturbations.

### B.4 Effect of using TRADES to train the teacher

We study the effect of using TRADES instead of adversarial training to train the teacher. In both cases, we employ the same training strategy as in Section 4. That is, using WideResNet-28 (He et al., 2016; Zagoruyko & Komodakis, 2016b) with swish activations (Hendrycks & Gimpel, 2016), model weight averaging (Izmailov et al., 2018) and additional data from Elucidating Diffusion Models (EDM)(Karras et al., 2022).

We observe in Table 11 the results of using TRADES instead of adversarial training. When using only the teacher to predict (i.e. no knowledge distillation), we see that TRADES is slightly more robust than adversarial training, but at a high cost in accuracy. Unfortunately, this decrease in accuracy is distilled to the student model as well when using AKD$_+$ or RSLAD$_+$. Moreover, it comes at no benefit in terms of robustness, underperforming when compared with using a teacher trained with adversarial training.

## C Teacher robustness trade-offs for AKD

### C.1 Ensemble of teachers performance

We study the effect of using an ensemble of teachers in Section 3.6, and show that they help improve the student performance. Here, we show that this increase in performance is not caused by the teacher itself being more robust or accurate. In fact, in Table 12 we show that using an ensemble of adversarially trained models does not improve the robustness or the clean accuracy compared to using one model. Strangely, when using AutoAttack to evaluate the ensemble, we find that the AutoAttack's black-box attacks were

Table 12: Comparison between the performance of one adversarially trained ResNet-18 model and an ensemble of four adversarially trained ResNet-18 models using different inititializations.

| Model | Clean | PGD-7 |
|---|---|---|
| ResNet-18 (AT) | 84.22 | 53.56 |
| Ensemble ResNet-18 (AT) | 81.86 | 52.74 |

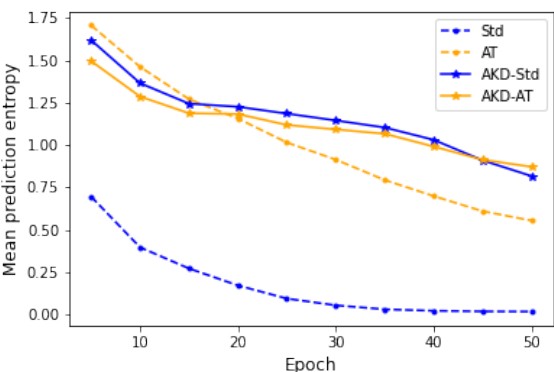

Figure 5: Mean entropy value of different optimization methods as the training progresses. The architecture is ResNet-18, used with the CIFAR10 dataset. AKD-Std and AKD-AT use AKD with a standardly trained and adversarially trained teacher, respectively.

ineffective. The ensemble reports an AutoAttack accuracy of 51.58% compared to the 46.99% obtained with a single model. Due to the small difference between the reported AutoAttack and PGD-7 accuracies of the ensemble, we believe further evaluation is required using adaptive black-box attacks that take the ensemble into account (He et al., 2017).

## C.2 Training trajectories

We finally provide an empirical study of the effect that AKD has on the functional properties of the student model. In particular, we find that using AKD modifies the calibration of the student model, mostly by modifying the training trajectory on hard-to-learn samples.

To test this behaviour, we first compare the evolution of the average entropy of the output probabilities of different models $H = \mathbb{E}_{\boldsymbol{x}}[-\sum_i f_S(\boldsymbol{x})_i \log f_S(\boldsymbol{x})_i]$, which measures how uncertain the model is on its predictions during training. We compute the probabilities on all the training data, and train using a ResNet-18 on CIFAR10. As we can see, AKD forces the student to learn solutions while being less over-confident than its teacher. This observation agrees with recent reports that suggest that CKD students are also better calibrated and less over-confident than standard models Müller et al. (2019).

It is widely known that deep neural networks do not use all training samples in the same way, and that certain training samples are harder to fit Arpit et al. (2017). What is more, in the context of adversarial robustness, it has been recently observed that the impact of harder-to-learn instances is even more noticeable than for standard models Sanyal et al. (2020); Liu et al. (2021); Dong et al. (2021). In this sense, it is expected that KD will have a stronger effect on those examples that can better exploit the *dark knowledge* of the teacher.

In order to test this hypothesis, we train our models on a binary-class dataset composed of all CIFAR10 samples that belong to the two first classes: *car* and *airplane*. Moreover, and following the practice in Liu et al. (2021); Dong et al. (2021), we rank the difficulty of learning each training and test sample based on

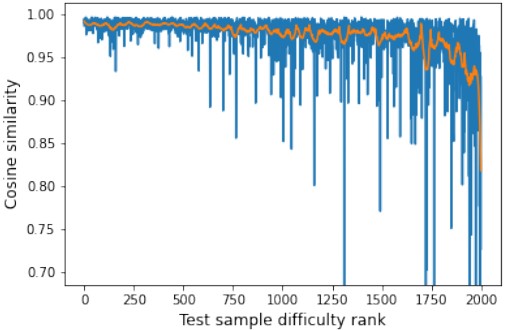 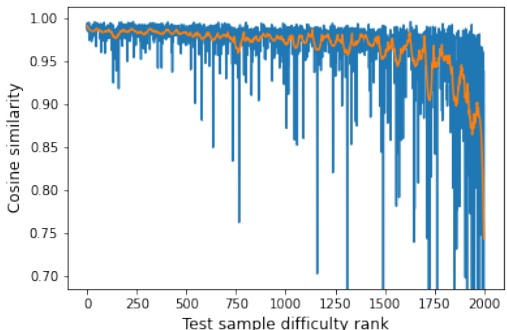

Figure 6: Cosine similarity when using AKD with an adversarially trained teacher when evaluated on natural (left) and adversarial (right) images. The blue line shows the true values, and the orange line a smoothed-out version.

the score

$$\mathcal{S}(\boldsymbol{x}) = \frac{1}{K} \sum_{k=1}^{K} \mathrm{CE}(f_k(\boldsymbol{x}), y), \tag{8}$$

where $f_k$ denotes the state of the neural network after $k$ epochs of training, and $K$ denotes the total number of training epochs.

Ranking the samples allows us to determine when does AKD improve performance. For each sample, we measure the improvement by recording the probability the model gives to the correct class[5] at the end of every epoch. We collect these values into a vector of probabilities $\boldsymbol{p}_S(\boldsymbol{x}) \in [0,1]^K$ in the case of the student, and $\boldsymbol{p}_T(\boldsymbol{x}) \in [0,1]^K$ in the case of the teacher. These vectors summarize the functional training trajectory of the models on a given sample $\boldsymbol{x}$, which means that their inner product measures the differences in their training dynamics.

Figure 6 shows precisely these differences on all natural and adversarial test samples, ordered in terms of learning difficulty[6]. As we can see, those samples which are easier to learn mostly follow the same trajectory on the student and teacher models, while the training trajectories clearly differ on hard examples, especially if they are adversarial. Interestingly, these differences in trajectory tend to have a strong calibration effect. As shown in Figure 7, the student model tends to assign higher probabilities on hard samples than the teacher, while it also tends to slightly reduce the confidence on the very easy examples. This effect is especially beneficial in terms of robustness, as it is consistent with the fact that difficult samples and label noise are more harmful to robustness than clean performance Sanyal et al. (2020); Liu et al. (2021); Dong et al. (2021). We believe this ability to improve generalization to difficult-to-learn regions can make KD very promising for multiple applications that require better performance in out-of-distribution data.

## D  Sample difficulty

In this Section we show some qualitative results that validate following the practice in Liu et al. (2021); Dong et al. (2021) to rank how difficult each sample is to learn. We apply the sample difficulty formulation equation 8 to a model standardly trained on the experimental settings we gave in Section C.2. We plot the 32 "easiest" and "hardest" images of planes and cars in Figures 8 and 9, respectively. Qualitatively, we find that the network distinguishes between car and plane images based on the overall color of the image and how busy is the background. An image with a lot of blue and light colors and simple sky backgrounds most probably is classified as a plane, while complex backgrounds with mostly red and dark colors are classified as car images. Thus, the network in general finds more difficulty on images that present uncommon features

---

[5]Well-calibrated models should assign high probabilities to the correct class, but not be too overconfident
[6]We provide results for a standardly trained teacher in the supplementary material

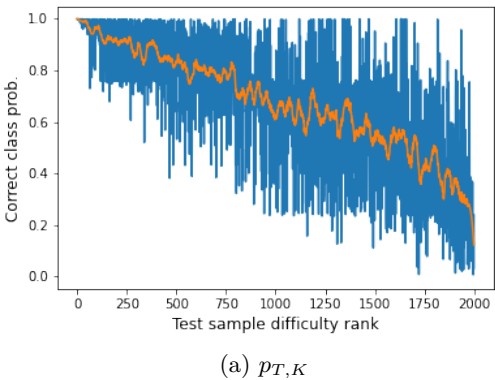
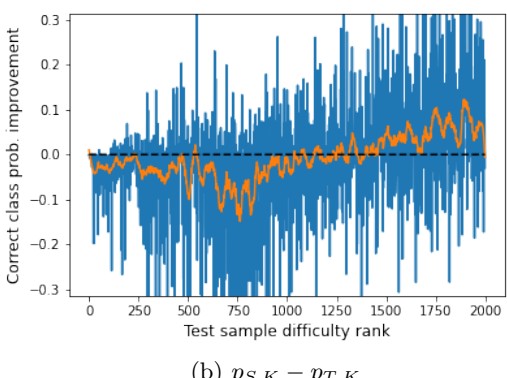

(a) $p_{T,K}$            (b) $p_{S,K} - p_{T,K}$

Figure 7: Correct class probability of the adversarially trained teacher (left) and the relative improvement of the student (right) tested on adversarial images. The blue line shows the true values, and the orange line a smoothed-out version.

or features from the other class. This is shown on our selected difficult images: planes with complicated backgrounds and cars with sky-like backgrounds.

# E  Training trajectories

We will now show extended results on the impact of AKD on the final performance and training trajectories of samples of different difficulty. The experimental settings are the same as the ones described in Section C.2.

## E.1  Adversarially trained teacher

We show in Section C.2 the calibration effect of AKD on the student model, increasing performance on hard samples at the cost of confidence on easier ones. Here, we show in Figure 10 that the effect AKD has on clean performance is quite similar. However, when compared with the results in terms of robustness we show in Figure 7, it is clear that AKD is more beneficial in terms of robustness, where more difficult samples improve and the improvement is more significant.

We also replicate this analysis using early-stopping and label mixing. We show in Figure 11a that early-stopping causes the accuracy to only increase for the most difficult samples, while reducing performance for most of them. As shown in Figure 11b, for robustness it is a bit less detrimental, but the effect is quite similar. We show in Figure 12 the same analysis, but for different values of the label mixing parameter $\alpha$. We find that label mixing increases clean and robust performance for all samples, but the effect increases proportionally with the sample difficulty. In this particular setting, using $\alpha = 0.3$ results in a optimum increase in performance.

## E.2  Standardly trained teacher

We show the difference between training dynamics between the teacher and the student, when using AKD with a standardly trained teacher. Figure 13 shows precisely these differences on all natural and adversarial test samples, ordered in terms of learning difficulty.

Compared with Figure 6, we see a similar behavior for the natural images. That is, the easier-to-learn samples mostly follow the same trajectory on the student and teacher models, while the training trajectories clearly differ on hard examples. When looking at the improvement on Figure 14, we see a large improvement for hard samples, at the cost of confidence on easier ones. Compared with Figure 10, the improvement is large thanks to the teacher being standardly trained, which transfer clean accuracy better than an adversarially trained teacher.

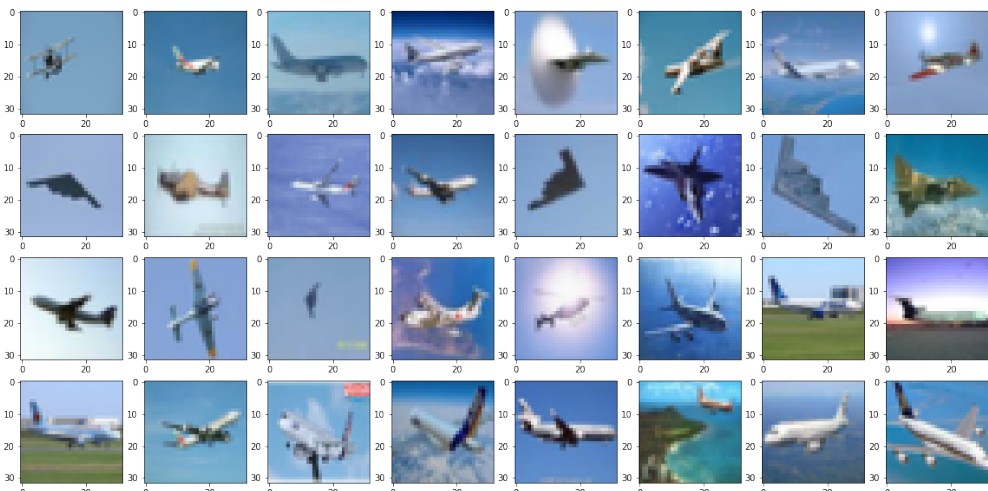

(a) Easiest to learn

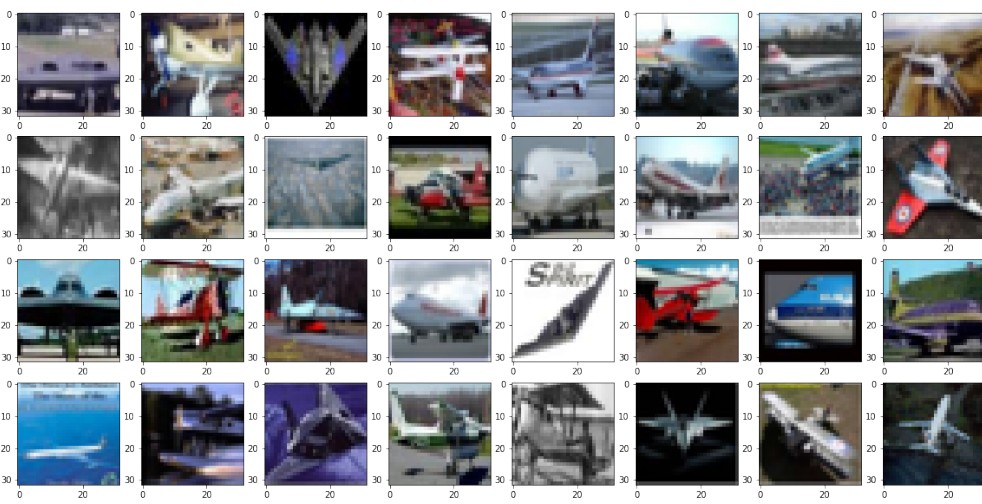

(b) Hardest to learn

Figure 8: Easiest and hardest to learn plane images based on the difficulty metric from Liu et al. (2021); Dong et al. (2021) applied to a standardly trained model on a subset of CIFAR10 containing only planes and cars.

However, in contrast, the adversarial images training trajectories change a lot in general, especially for hard samples (see Figure 13). This can be explained by the settings of the experiments. Since we are evaluating the effect of AKD in itself, we do not use any label mixing. We find that the difference in trajectories is slightly correlated with the teacher performance. Thus, the trajectories will be more similar for the easier samples, which the teacher classifies correctly due to the simplicity of the task (see Figure 15). But as the samples are harder, the robustness of the teacher drops, with makes the trajectories also more dissimilar.

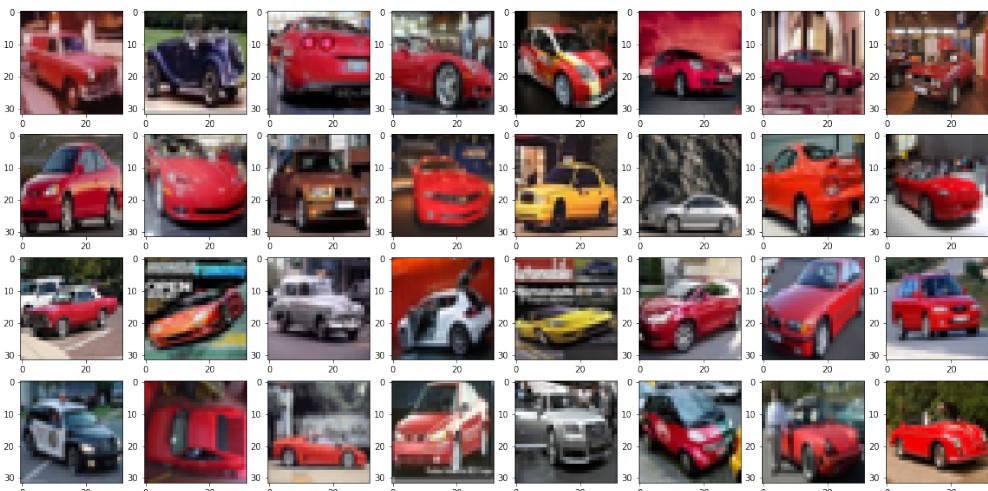

(a) Easiest to learn

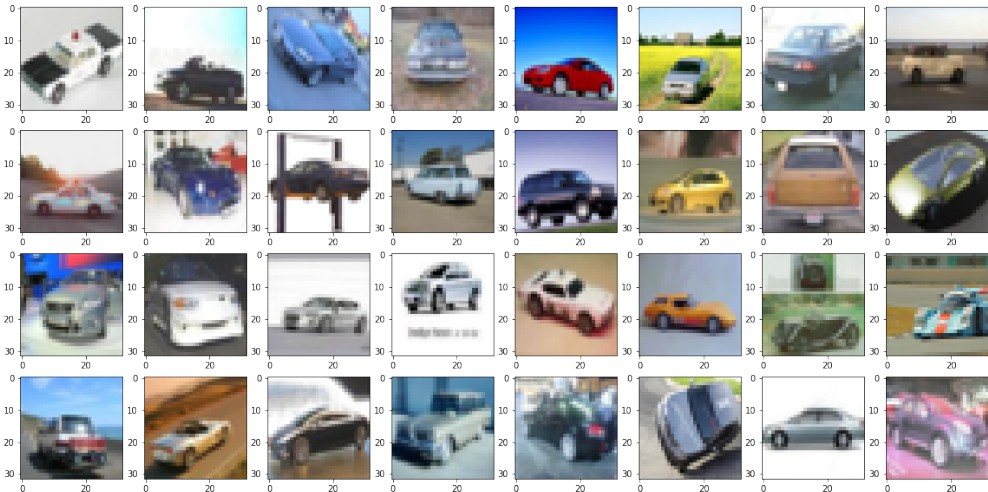

(b) Hardest to learn

Figure 9: Easiest and hardest to learn car images based on the difficulty metric from Liu et al. (2021); Dong et al. (2021) applied to a standardly trained model on a subset of CIFAR10 containing only planes and cars.

We can see in Figure 15 that AKD helps increase slightly the confidence for all samples, but it does not correct misclassification by itself.

We also replicate this analysis using early-stopping and label mixing. We show in Figure 16a that early-stopping reduces the accuracy for hard samples. This reduction is consistent with Cho & Hariharan (2019), where they claim that for CKD, early stopping does not help improve clean performance when there is no capacity mismatch between the teacher and student models. In contrast, robustness increases for easier samples, as shown in Figure 11b.

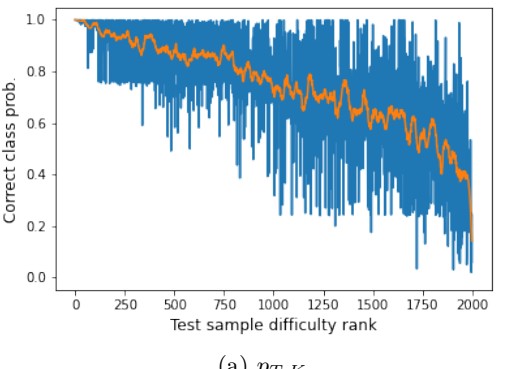 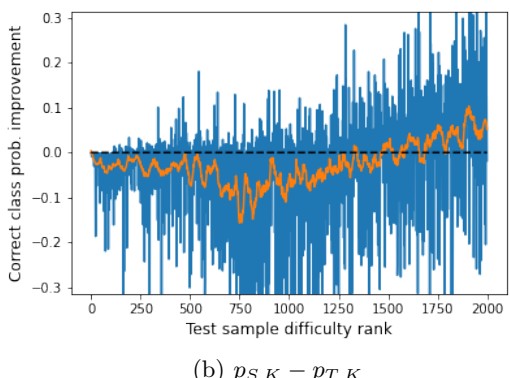

(a) $p_{T,K}$

(b) $p_{S,K} - p_{T,K}$

Figure 10: Correct class probability of the adversarially trained teacher (left) and the relative improvement of the student (right) tested on natural images. The blue line shows the true values, and the orange line a smoothed-out version.

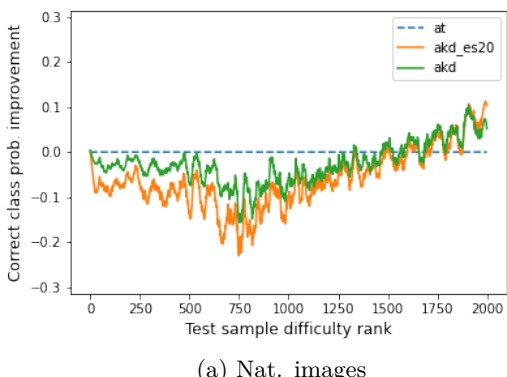 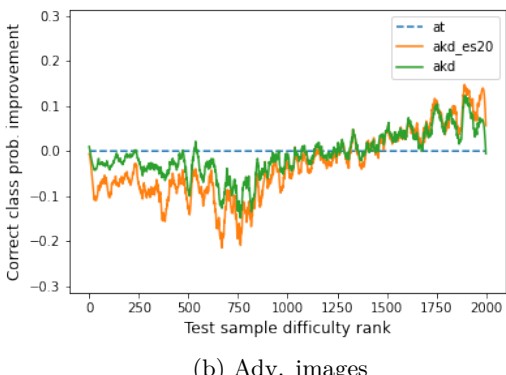

(a) Nat. images

(b) Adv. images

Figure 11: Smoothed improvement of the correct class probability $(p_{S,K} - p_{T,K})$ when using AKD with and without early stopping an adversarially trained teacher. We evaluate the performance on natural images (left) and adversarial ones (right).

We show in Figure 12 the same analysis, but for different values of the label mixing parameter $\alpha$. We find that label mixing increases clean performance for all samples, but the effect increases proportionally with the sample difficulty. In terms of robust performance, we see the same behavior we presented in Figure 1 when using a standardly trained model. That is, for high values of $\alpha$, the student is not robust, while for small values the robustness is high. This analysis shows it is a bit more nuanced than the previously presented results, and for high values of $\alpha$ the model improves a bit its confidence in the correct class, but not enough to avoid misclassification.

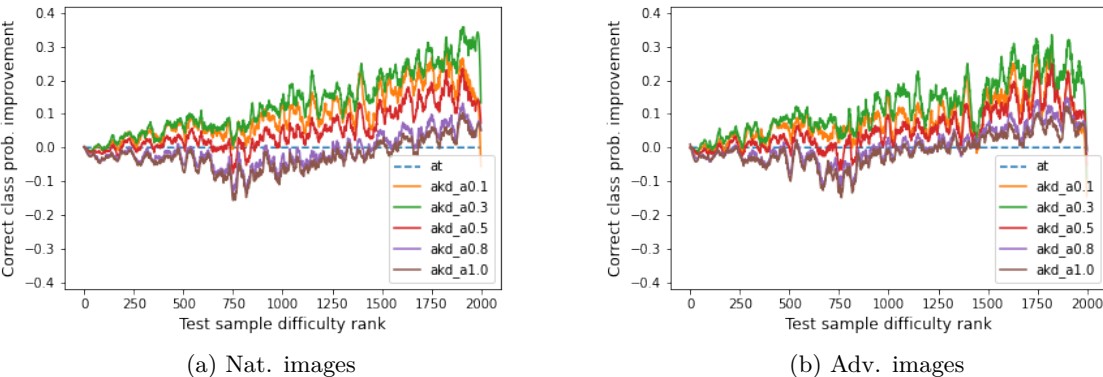

(a) Nat. images

(b) Adv. images

Figure 12: Smoothed improvement of the correct class probability $(p_{S,K} - p_{T,K})$ when using AKD with an adversarially trained teacher, for different values of the label mixing parameter $\alpha$. We evaluate the performance on natural images (left) and adversarial ones (right).

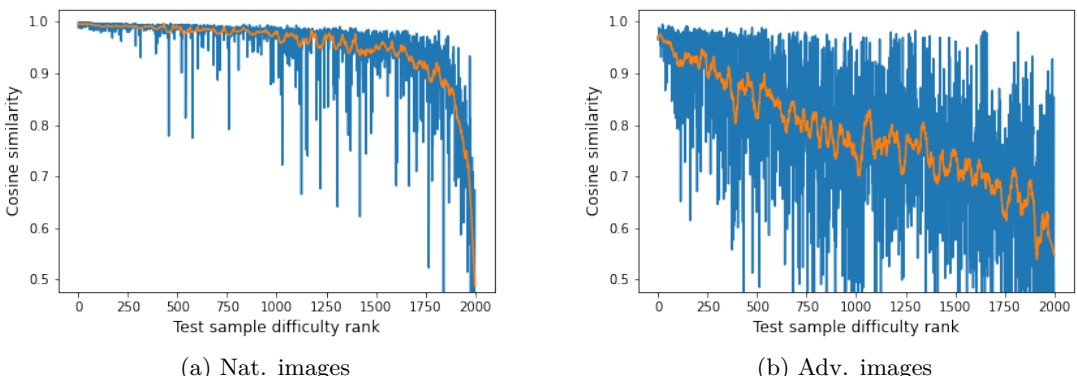

(a) Nat. images

(b) Adv. images

Figure 13: Cosine similarity when using AKD with an standardly trained teacher when evaluated on natural (left) and adversarial (right) images. The blue line shows the true values, and the orange line a smoothed-out version.

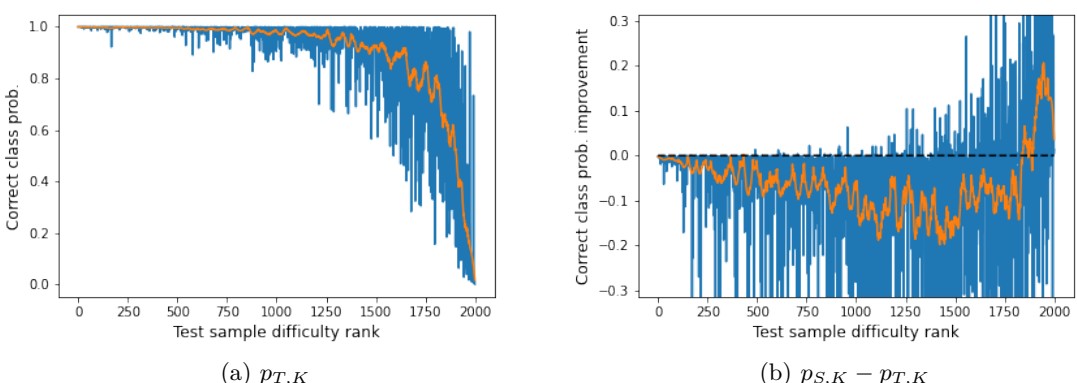

(a) $p_{T,K}$

(b) $p_{S,K} - p_{T,K}$

Figure 14: Correct class probability of the standardly trained teacher (left) and the relative improvement of the student (right) tested on natural images. The blue line shows the true values, and the orange line a smoothed-out version.

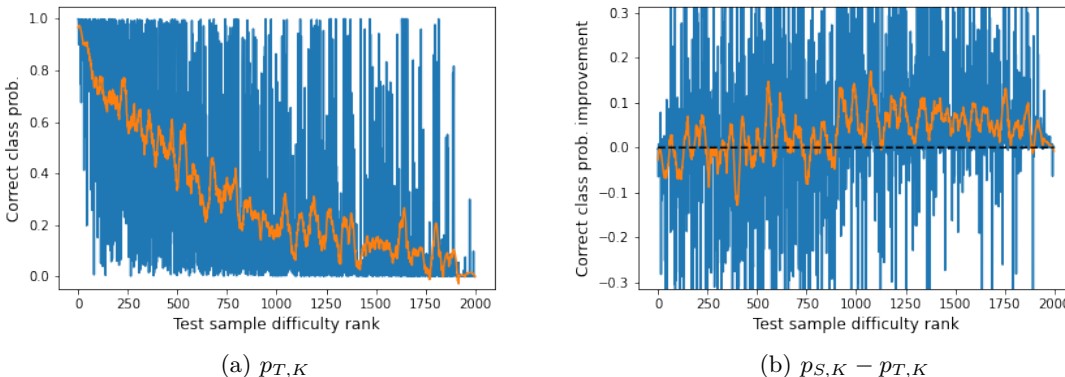

(a) $p_{T,K}$  (b) $p_{S,K} - p_{T,K}$

Figure 15: Correct class probability of the standardly trained teacher (left) and the relative improvement of the student (right) tested on adversarial images. The blue line shows the true values, and the orange line a smoothed-out version.

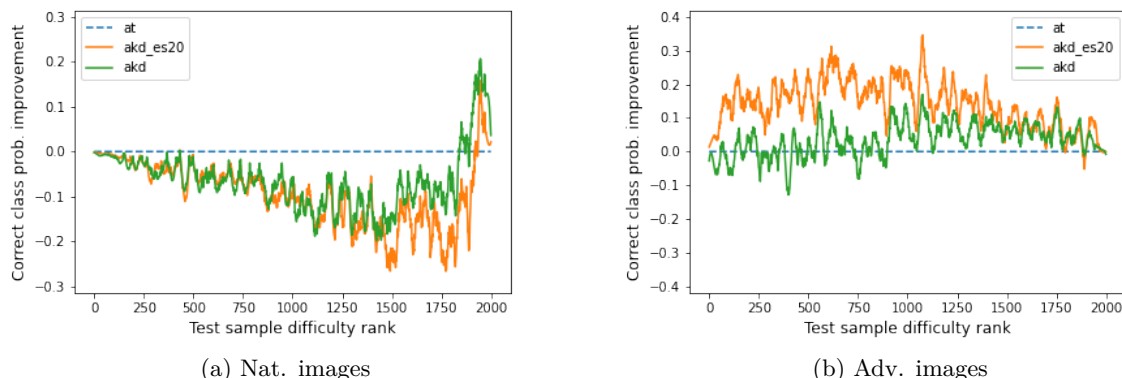

(a) Nat. images  (b) Adv. images

Figure 16: Smoothed improvement of the correct class probability ($p_{S,K} - p_{T,K}$) when using AKD with and without early stopping a standardly trained teacher. We evaluate the performance on natural images (left) and adversarial ones (right).

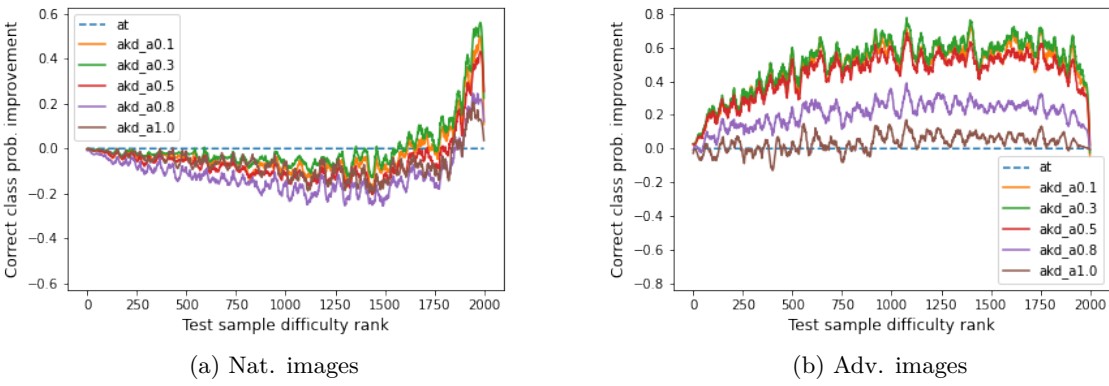

(a) Nat. images  (b) Adv. images

Figure 17: Smoothed improvement of the correct class probability ($p_{S,K} - p_{T,K}$) when using AKD with a standardly trained teacher, for different values of the label mixing parameter $\alpha$. We evaluate the performance on natural images (left) and adversarial ones (right).

