# OpenReview forum: "When does knowledge distillation improve robustness?"
_TMLR — Rejected by TMLR_

### Review · Reviewer_WfHu · 2023-07-06

**Summary Of Contributions:**

This paper provides an empirical study of the knowledge distillation for adversarial robustness. Specifically, the authors show that prior objective functions cannot achieve satisfactory robustness after knowledge distillation. Instead, the authors propose to minimize the gap between students and teachers in the adversarial regions. In the experiments, the authors also provide several discoveries to reveal the optimal setting for adversarial knowledge distillation.

**Audience:**

Yes

**Broader Impact Concerns:**

No broader impact concerns.

**Claims And Evidence:**

Yes

**Requested Changes:**

1. Please clarify the novelty to highlight the contribution compared with prior work. Some theoretical analysis of AKD is preferred to reveal some insights.
2. Please include more evaluation large scale datasets.
3. Please include more evaluation with advanced adversarial training methods.
4. Please compare with recent baselines.


**Strengths And Weaknesses:**

Strengths:
1. The paper is well-written and easy to follow.
2. The authors conduct extensive ablation studies, including early stopping, distillation from standard or adversarial trained teachers, and distillation from model ensemble.
3. The empirical observation and discoveries could benefit the community.

Weaknesses:

I have several concerns:
1. The novelty is limited. The proposed AKD and label mixing seem some naïve baselines. The technical contribution is trivial compared with prior work.
2. Lack of evaluation on large scale datasets, such as Tiny-ImageNet or ImageNet.
3. Although the authors include many ablation studies, the evaluation with different advanced adversarial training methods is missing, such as MART, TRADES, etc.
4. The baselines are out-of-date. Some advanced adversarial knowledge distillation approaches have been introduced, which should be discussed and compared [1].

[1]. Boosting Accuracy and Robustness of Student Models via Adaptive Adversarial Distillation.

---

> ### Author Response · Authors · 2023-07-25
>
> We appreciate that the reviewer points that the paper empirical analysis is useful to the community, that the experiments are extensive and comprehensive. Regarding the requested changes:
> - We would like to clarify that the main objective of our paper is to provide a “thorough analysis of different robust knowledge distillation (RKD) techniques with the aim to provide general guidelines to improve the adversarial performance of a student model.” In this regard, AKD is just another loss we test in our ablations. We never claim that AKD is uniformly superior to any other loss, and instead show that the best choice of loss is nuanced. For example, as shown in Table 6 AKD performs better than other losses CIFAR10 augmented with DDPM samples, while RSLAD+ obtains better robustness in vanilla CIFAR10, as shown in Table 4. Our main motivation for introducing AKD comes from a void we noticed in the RKD literature which had not been updated in light of the most recent results in the clean knowledge distillation (CKD) field. Namely, it has been consistently observed that objectives that try to match the functions of the student and the teacher outperform other losses in CKD [1]. For this reason, in our work we introduced AKD to better mimic and study that idea in an adversarial setting. However, we have edited the paper to motivate the design choice that is specific to AKD, and not RSLAD+ or ARD+: function matching on the adversarial region.
> - We have performed all our experiments on CIFAR10 and CIFAR100 because they are the main benchmarks used to test robustness in the literature. As mentioned by the reviewers, our experiments are thorough and test a wide range of conditions. However, after the reviewer comments we have included experiments on Tiny-ImageNet in Appendix B.2. Our main conclusion holds: distillation can be used to improve the trade-off between robustness and accuracy. We remark however that due to the lower achievable robustness on Tiny-ImageNet with adversarial training, agreeing with our insights, the improvements in robustness on this dataset are minimal. We see a clear improvement in clean accuracy, otherwise. The similarity with the results obtained in Table 2 for lower capacity models, makes us think that for larger networks, knowledge distillation could be effective on Tiny-ImageNet. Finally, we note that performing adversarial training experiments on ImageNet is clearly beyond what our computational budget allows, and is not the standard in the adversarial robustness literature due to its high cost.
> - We agree that results on more advanced adversarial training methods could better contextualize our results, as it provides more baselines. However, the focus of our paper is on how these advanced adversarial training methods may be improved with knowledge distillation. That is why we made experiments with the best performing methodology for robustness, based on robustbench benchmark results, and used the robust knowledge distillation in conjunction with these techniques. Since TRADES and MART are loss functions, they cannot be used as is to distill knowledge to a student. In fact, one could claim that ARD is the modified version of TRADES that distills the teacher predictions. While finding the modified version of MART would be an interesting endeavor, the focus of our paper was not to find the best loss function, but to provide an analysis on how different factors like early-stopping, label mixing, model capacity, or even the loss function used can affect the performance of knowledge distillation. Nonetheless, we included the results of using TRADES instead of adversarial training to train the teacherin Appendix B.4, which actually results in worse student performance than using AT for the teacher.
> - Regarding comparison with recent baselines, we found that most of them either are concurrent with our work, like the paper cited by the reviewer, which was published on June 18th, one week before we submitted this paper, or they cite our arxiv paper, which would compromise the double-blind nature of the review process. We think our methodology can be applied to these more recent adversarial training methods, in the same way we could apply it to the other robust knowledge distillation methods.

---

### Review · Reviewer_2bMA · 2023-07-14

**Summary Of Contributions:**

This paper shows a comprehensive empirical study of 3 adversarially knowledge distillation methods. The author proposed a new method called AKD, which replaces the KL loss in ARD with the CE loss. They did comprehensive experiments on tuning the hyper-parameters in these methods where the findings are inspiring and useful in practice.

**Audience:**

Yes

**Broader Impact Concerns:**

No concerns.

**Claims And Evidence:**

Yes

**Requested Changes:**

1. Explain the motivation of their proposed method. Show evidence that their method outperforms the other baselines and give explicit analysis on this.
2. Add an explanation in the caption for the "Layer size" in Table 2.
3. Highlight the key conclusions in the analysis.

**Strengths And Weaknesses:**

Strengths: the experiments are comprehensive in terms of the effects of different hyper-parameters and the suggestions on how to choose these parameters are useful in practice.


The main weakness is their proposed methods doesn't seem to outperform the baseline. It also lacks a motivation and an explanation about why they think CE would outperform KL and thus replacing it. The experimental results seem to show the baseline ARD (uses KL) outperforms the proposed method (uses CE).

Questions for the authors:

1. The paper proposes to align the student's and the teacher's outputs using cross-entropy. However, the authors didn't provide any reasoning about why they change the generally applied KL divergence to the CE loss. On one side, it lacks explanation on the change. On the other size, only changing KL loss to CE loss doesn't count towards novelty.
2. In Figure 1, it seems the ARD methods with accurate teacher performs the best. Is it a support saying that there's no need to change KL to CE?
3. In Figure 2, it still seems the ARD+ has the best performance over the proposed AKD method. While Figure 1 shows alpha doesn't affect the performance of ARD, which means the ARD is still the best method overall?
Why Table 4 and Table 6 doesn't have ARD result for comparison?
Minor suggestions:

---

> ### Author Response · Authors · 2023-07-25
>
> We appreciate that the reviewer points that the paper suggestions are useful, and that the experiments are comprehensive. Regarding the questions:
> - We would like to clarify that the main objective of our paper is to provide a “thorough analysis of different robust knowledge distillation (RKD) techniques with the aim to provide general guidelines to improve the adversarial performance of a student model.” In this regard, AKD is just another loss we test in our ablations. We never claim that AKD is uniformly superior to any other loss, and instead show that the best choice of loss is nuanced. For example, as shown in Table 6 AKD performs better than other losses CIFAR10 augmented with DDPM samples, while RSLAD+ obtains better robustness in vanilla CIFAR10, as shown in Table 4. We use the CE loss in AKD inspired by vanilla adversarial training. As the reviewer remarks, other methods like ARD, are instead inspired by TRADES, which combines CE loss in the natural examples with KL loss in the adversarial region.
> - The observations we obtain in this paper are nuanced, and we show that no loss function outperforms all others in all scenarios. One of the main objectives of this work is to help the community navigate those nuances. While ARD seems to outperform the other losses in the main analysis presented with ResNet-18 and vanilla CIFAR10, we would like to point out that in this same scenario we can obtain better robustness by using the RSLAD loss function with robust teacher instead, at the cost of some accuracy.
> - Regarding the choice of CE loss instead of KL loss in AKD, we reiterate our answer to the first question. Our objective with this paper is not to introduce and motivate a new loss function, rather it is an ablation study with the clear goal of understanding to what extent different RKD techniques can be used to improve the robustness and accuracy of adversarially trained networks. Thus, we think the paper should not be judged based on the design choices of the loss functions. We have reworded different sections of the paper to make sure it does not give the impression that we are proposing a single new RKD method. However, as suggested by the reviewer, and to improve the ablation study, we added the performance of ARD on Table 6. Compared with the previous scenario described in the previous question, we see that in that case it underperforms compared to AKD and RSLAD. We conjecture that the reason AKD outperforms ARD in Table 6 is that function matching in the adversarial region could be more impactful for bigger models, which are more prone to overfitting the natural samples.
>
> Regarding the requested changes:
> - Our objective with this paper is to study in which conditions knowledge distillation can be most effective, not necessarily to promote the AKD loss function over the other robust knowledge distillation methods presented. Our main contribution is the analysis that we presented on when all robust knowledge distillation methods can be most effective (i.e. early stopping, label mixing, the teacher and student model capacity, ensembling, additional training data, …), where the loss function is just one of the variables. We do not claim our method is better in general, and in fact ARD+ outperforms it on the ResNet18 experiments, but we show that it is most effective for bigger architectures on the state-of-the-art adversarial training methodology. However, we have edited the paper to better motivate the design choice that is specific to AKD, and not RSLAD+ or ARD+: function matching on the adversarial region. And, as said before, we have reworded different sections of the paper to make sure it does not give the impression that we are proposing a single new RKD method.
> - We have edited the paper to include in the Table 2 caption the explanation provided in the main text.
> - We believe we have already highlighted the key conclusions of our analysis in the last paragraph of Section 3.

---

### Review · Reviewer_xmnv · 2023-07-20

**Summary Of Contributions:**

This paper focuses on adversarial robustness in the framework of knowledge distillation. To sum up, this paper demonstrates the importance of early stopping, model ensembling, label mixing, and the weakly adversarially trained teacher model for improving the student model's performance under the framework. Specifically, they introduce a new adversarial knowledge distillation loss, namely AKD, to match the outputs in the adversarial regions. Comprehensive experiments about those robust knowledge distillation methods show that the performance gain is related to the data quality of training.

**Audience:**

Yes

**Claims And Evidence:**

Yes

**Requested Changes:**

Larger scale experiments; More comprehensive evaluation on robust accuracy; Enhance the motivation part for AKD and explain the rationality of using AKD with the adversarial outputs of the teacher models.

**Strengths And Weaknesses:**

Strengths:
1. This paper studies another important objective, adversarial robustness, under the knowledge distillation framework.
2. The empirical insights extracted from those comprehensive experiments demonstrated that early stopping, label mixing, and model ensembling are important for benefiting the student model's adversarial robustness, which is useful for future in-depth research.
3. Comprehensive comparisons among different variants of robust knowledge distillation methods are conducted to provide empirical support.

Weaknesses:
1. The experiments only consider those small-scaled datasets like CIFAR-10/100, which shows limited information on whether those empirical findings are still valid when we trained on more large datasets like ImageNet.
2. Since most empirical findings are supported by the experimental results about performance change, the evaluation metrics of robust accuracy can be more comprehensive, like including more different kinds of attacks.
3. The significant difference of AKD from the previous variants of RKD is using the adversarial outputs of the teacher model, it is better to discuss the motivation of such a design clearly.

---

> ### Author Response · Authors · 2023-07-25
>
> We appreciate that the reviewer points that the paper studies an important objective, and that our experiments and comparisons between robust knowledge distillation methods are comprehensive and useful for future research. Regarding the requested changes:
> - We have performed all our experiments on CIFAR10 and CIFAR100 because they are the main benchmarks used to test robustness in the literature. As mentioned by the reviewers, our experiments are thorough and test a wide range of conditions. However, after the reviewer comments we have included experiments on Tiny-ImageNet in Appendix B.2. Our main conclusion holds: distillation can be used to improve the trade-off between robustness and accuracy. We remark however that due to the lower achievable robustness on Tiny-ImageNet with adversarial training, agreeing with our insights, the improvements in robustness on this dataset are minimal. We see a clear improvement in clean accuracy, otherwise. The similarity with the results obtained in Table 2 for lower capacity models, makes us think that for larger networks, knowledge distillation could be effective on Tiny-ImageNet. Finally, we note that performing adversarial training experiments on ImageNet is clearly beyond what our computational budget allows, and is not the standard in the adversarial robustness literature due to its high cost.
> - We have followed the standard practice in the field and use AutoAttack to measure robustness, as this is the de facto tool agreed by the community to evaluate robustness. AutoAttack is a battery of strong adversarial attacks including SOTA white-box and black-box adversarial attacks. In this regard, we believe our evaluation is solid as it covers the most important axis of an adversarial robustness evaluation.
> - We would like to clarify that the main objective of our paper is to provide a “thorough analysis of different robust knowledge distillation (RKD) techniques with the aim to provide general guidelines to improve the adversarial performance of a student model.” In this regard, AKD is just another loss we test in our ablations. We never claim that AKD is uniformly superior to any other loss, and instead show that the best choice of loss is nuanced. For example, as shown in Table 6 AKD performs better than other losses CIFAR10 augmented with DDPM samples, while RSLAD+ obtains better robustness in vanilla CIFAR10, as shown in Table 4. Our main motivation for introducing AKD comes from a void we noticed in the RKD literature which had not been updated in light of the most recent results in the clean knowledge distillation (CKD) field. Namely, it has been consistently observed that objectives that try to match the functions of the student and the teacher outperform other losses in CKD [1]. For this reason, in our work we introduced AKD to better mimic and study that idea in an adversarial setting. After the reviewer’s comment, we have motivated AKD further in the new version of our manuscript.
>
> [1] : Lucas Beyer, Xiaohua Zhai, Amélie Royer, Larisa Markeeva, Rohan Anil, and Alexander Kolesnikov. Knowledge distillation: A good teacher is patient and consistent. p. 10925–10934, 2022

---

### Review · Reviewer_t3Lw · 2023-08-05

**Summary Of Contributions:**

This paper analyzes different perspectives to measure the ability of robust knowledge distillation (RKD) to improve model robustness. Specifically, the paper investigates the effect of early stopping on RKD when training a teacher model. It also analyzes the effect of label mixing on RKD. In addition, the effect of different teacher training on the accuracy and robustness of student models is outlined. Finally, a new loss function called Adversarial Knowledge Distillation (AKD) is proposed.


**Audience:**

Yes

**Claims And Evidence:**

Yes

**Requested Changes:**

The dataset used in the experiments is a low-resolution single object image recognition dataset with at most 200 classes, and only resnet-18 and wideresnet-28 networks are used, which raises questions about their generalization performance. The reviewer would like to know where the guarantee of the validity of the guideline comes from, other than the experimental setting used.

In relation to the first comment, the conclusions drawn are general ones regarding RKD. It would be necessary to clarify the scope, clearly state the limitations, and avoid overstating the findings of this paper.


**Strengths And Weaknesses:**

Strengths
- In this paper, a thorough analysis of how the RKD method can be used to improve the robustness of the model was conducted.
- A new loss function called Adversarial Knowledge Distillation (AKD) was proposed.
- Recipes were given for using early stopping, label mixing, ensemble, and adversarial learning to improve the performance of the student models.
- This paper is well written and easy to read.

Weaknesses
- In this paper, various experiments have tested how early stopping, label mixing, ensembles, and adversarial learning should be used to perform robust knowledge distillation. However, it is difficult to guarantee the generalization of the obtained results.
- The main datasets considered are cifar10 and cifar100, and this paper presents guidelines and concludes with results obtained only on small object recognition datasets. In the appendix, this paper presents experiments using a tiny ImageNet, but in each case the results are obtained only for object recognition with a single object in a small image size, but general results on robust knowledge distillation are derived from these results.
- The network architectures used are only resnet-18 and wideresnet-28, and this paper has not experimented with different architectures such as recent transformer-based architectures, making it difficult to verify the generalization performance.

---

> ### Author Response · Authors · 2023-08-14
>
> We appreciate that the reviewer points to the thoroughness of the analysis proposed, and that the experiments and methodology proposed is comprehensive. Regarding the requested changes:
> - ​​As mentioned by the reviewers, we believe the analysis is thorough in the settings studied and we were able to draw conclusions and elucidate how the different factors studied (e.g. label mixing) affect the performance of robust knowledge distillation. Possibly more experiments on different tasks (e.g. image segmentation) would be interesting, however we focused on studying image classification as it is the most commonly used framework for robustness.
> - We have performed all our experiments on CIFAR10 and CIFAR100 because they are the main benchmarks used to test robustness in the literature. There are two main factors that limit how varied are the existing benchmarks. First, the increased computational cost of adversarial training and computing adversarial perturbations, especially AutoAttack for testing, which is required for proper robustness evaluation but takes more than a thousand times the time of a regular forward pass. For clarity, each training iteration takes around 10x times the standard training time for all the methods that we benchmark, and memory consumption is at least 2x higher since we have to store the gradients for adversarial computation and the additional teacher model. Moreover, convergence in RKD methods takes more epochs and data to achieve comparable results to standard training. Second, the complexity of the robustness objective, where even for relatively simple recognition tasks like the ones presented, where there is a single object in a relatively small image, the achievable robustness is quite low compared with the achievable accuracy. In fact, we showed with the experiment of Tiny-Imagenet, that robust knowledge distillation is not viable because of the low teacher robustness. We expect that this result would be similar for even more complex tasks, for which the teacher robustness is expected to be even lower.
> - We understand that recently large architectures like ViT’s have become widely popular and we acknowledge the value of the reviewer’s suggestion. However, given the additional experiments would be quite expensive, we politely ask if these experiments would be essential for the publication of the manuscript. As mentioned, our primary aim is to elucidate how the different factors studied (e.g. early stopping) affect the performance of robust knowledge distillation. With this objective, we focused on the most extensively examined fast adversarial training settings, which, to the best of our knowledge, have predominantly been proposed for classical CNNs. For that purpose, we presented results with the two most popular architectures within this scope (ResNet-18, WideResNet-28) and corroborated our experiments with multiple random seeds and multiple ablations of these factors. Even when benefitting from the available well-tuned hyperparameter to perform adversarial training on these architectures, this resulted in more than 10,000 GPU hours. Integrating ViT or other larger architectures would require an even larger amount of compute. It would also require significant effort, since the effective usage of ViT in robustness settings is still an ongoing area of research [1,2].
>
> [1] Liu, Chang, et al. "A comprehensive study on robustness of image classification models: Benchmarking and rethinking." arXiv 2023.
>
> [2] Debenedetti, Edoardo, Vikash Sehwag, and Prateek Mittal. "A light recipe to train robust vision transformers." IEEE SaTML 2023.

---

### Author Response · Authors · 2023-07-25

We thank all the reviewers for their valuable feedback. We really appreciate all the contributions and value all the effort and time invested to help us refine our paper. We are grateful for all the encouraging comments and thoughtful requests. We would like to use this comment to summarize all the changes that we made to our paper (marked in red) based on your feedback:
- Added results with ARD/ARD+ method in Table 6
- Improved clarity of caption in Table 2
- Added experiments with Tiny-ImageNet in Appendix B.2
- Added results of using TRADES with the teacher in the Appendix B.4
- Motivated why we use function matching in AKD
- Reworded text to not give the impression that our objective is to motivate the AKD loss function

---

### Decision · Action_Editors · 2023-08-29

**Recommendation:** Reject

**Comment:**

The recommendation is based on the reviewers' comments, the editor's personal evaluation, and the post-rebuttal discussion.

While all reviewers see some merits and technical novelty in the proposal of studying distillation and robustness, there are several remaining concerns about the technical contributions and the validity of the claimed results, including

1. Lack of direct and convincing evidence on how the robustness claims from the current results can generalize to other setups, and why can they generalize? In particular, many reviewers are concerned that the dataset tested is mainly a small object recognition dataset, and the networks used are only resnet-based networks.

2. The evaluation metrics of the robust accuracy also include different dimensions like adversarially attack methods, adversarially attack strength, and adversarial constraint on samples. The AA attack mentioned in the authors' response is just about changing the specific methods for evaluation but not including other dimensions of information for thoroughly evaluating the performance of the robust student model.

3. Since the target of this research work is exploring when knowledge distillation improves robustness, it would be better if the authors could provide more general guidance on the research direction with new insight or new algorithmic design. The current version of the work can be better improved if any theoretical insights can be drawn from those empirical experiments; it would be more convincing and generalizable for the obtained results.

The reviewers actively engaged during the discussion phase, while the authors' rebuttal did not change the final assessment of the current work. Therefore, this work should be revised significantly and undergo another round of full reviews. I hope the reviewers’ comments can help the authors prepare a better version of this submission.

During the discussion, no reviewers were willing to champion this submission. I hope the reviewers’ comments can help the authors prepare a better version of this submission. I also encourage the authors to revise and resubmit this paper to TMLR, with the aim of widening and sharpening the claims and findings.

**Audience:**

Yes. This topic (distillation and robustness) is of interest to TMLR's audience

**Claims And Evidence:**

Several reviewers pointed out that the claims are validated only on a limited set of tasks with insufficient performance improvement, and may lack generalization.

**Resubmission Of Major Revision:**

The authors may consider submitting a major revision at a later time.